# Human observers have optimal introspective access to perceptual processes even for visually masked stimuli

Megan A K Peters[1]*, Hakwan Lau[1,2]

[1]Department of Psychology, University of California, Los Angeles, Los Angeles, United States; [2]Brain Research Institute, University of California, Los Angeles, Los Angeles, United States

**Abstract** Many believe that humans can 'perceive unconsciously' – that for weak stimuli, briefly presented and masked, above-chance discrimination is possible without awareness. Interestingly, an online survey reveals that most experts in the field recognize the lack of convincing evidence for this phenomenon, and yet they persist in this belief. Using a recently developed bias-free experimental procedure for measuring subjective introspection (confidence), we found no evidence for unconscious perception; participants' behavior matched that of a Bayesian ideal observer, even though the stimuli were visually masked. This surprising finding suggests that the thresholds for subjective awareness and objective discrimination are *effectively the same*: if objective task performance is above chance, there is likely conscious experience. These findings shed new light on decades-old methodological issues regarding what it takes to consider a neurobiological or behavioral effect to be 'unconscious,' and provide a platform for rigorously investigating unconscious perception in future studies.

*For correspondence:
meganakpeters@ucla.edu

**Competing interests:** The authors declare that no competing interests exist.

## Introduction

Above-chance performance without awareness in perceptual discrimination tasks is a strong form of unconscious perception. In these demonstrations (e.g., blindsight: *Weiskrantz, 1986*) the subjective threshold for awareness (when a stimulus is consciously 'seen') seems well above the objective threshold for forced-choice discrimination (when a stimulus can be correctly identified): subjects can discriminate a target above chance performance, yet report no awareness of the target. Many researchers believe normal, healthy subjects can also directly discriminate near-threshold, low-intensity targets without subjective awareness (e.g., *Boyer et al., 2005*; *Charles et al., 2013*; *Merikle et al., 2001*; but see *Snodgrass et al., 2004* for an opposing view).

We conducted an informal survey to confirm this popular belief, which also revealed that many believe convincing evidence for this phenomenon is lacking. We asked survey participants three key questions: (1) "Do you believe in subliminal perception?" (2) "Do you believe that the subjective threshold for awareness is above the objective discrimination threshold?" and (3) "If 'yes', do you believe this has been convincingly demonstrated in the literature?" Most respondents reported believing that subliminal processing exists (94%), but also that they did not believe it had been convincingly demonstrated in the literature (64%). These belief patterns were shown even among those who reported having published on subliminal or unconscious perception (94% and 61%, respectively). See *Appendix 1* for full text of questions and detailed survey results.

A primary culprit in this controversy is the problem of criterion bias: an observer's report of 'unseen' doesn't necessarily imply *complete* lack of awareness, only that the stimulus' strength fell below some arbitrary boundary for reporting 'seen' (*Eriksen, 1960*; *Hannula et al., 2005*;

**eLife digest** In the 1980s, psychologists made an unexpected discovery while working with individuals who had become blind after sustaining damage to areas of the brain required for vision. These individuals could respond correctly to questions about the shape and location of objects in their visual field, even though they could no longer see the objects. This phenomenon became known as 'blindsight', and it is regarded as a classic example of perception in the absence of conscious awareness.

Many researchers who study consciousness believe that everyone is capable of subliminal or unconscious perception: that is, of detecting and processing stimuli without being consciously aware of them. However, studies investigating this phenomenon have produced contradictory results. Peters and Lau have now tested unconscious perception directly, using a recently developed method that overcomes some of the problems faced by previous studies.

Human volunteers took part in several trials, in which they were shown two images. Each image was 'masked' to prevent the volunteers from consciously registering them. After each image was shown, the volunteers had to state whether a patch of gray and white stripes in the masked image was tilted to the left or to the right. However, one of the two images did not include a gray and white patch. After seeing both images in a trial, the volunteers also had to indicate which of their answers they were most confident about.

If the volunteers could perceive the patches without being consciously aware of doing so, their response should show two features. The volunteers should correctly state the tilt direction of the stripes more often than would be expected if they were guessing at random. However, they should also feel no more confident in their responses for the images that did feature a striped patch than for the 'no patch' ones. Peters and Lau found no such evidence of unconscious perception.

Nevertheless, the volunteers were consistently better at correctly stating the direction the stripes were tilted in than their confidence ratings would suggest. Does this indicate some degree of perception without awareness? Peters and Lau argue that it does not, because a computer model designed to perform the task showed a similar level of performance to the volunteers.

These findings suggest that previous reports of unconscious perception may have been contaminated by the problems that Peters and Lau controlled for, and that perhaps unconscious perception doesn't occur in people without brain damage. Researchers will now need to do more studies using similar approaches to determine whether observers without brain damage can truly experience unconscious perception, and how such unconscious perception might be represented in the brain.

*Lloyd et al., 2013*; *Merikle et al., 2001*). Unfortunately, most methods of studying unconscious perception suffer from this 'criterion problem' (e.g., *Charles et al., 2013*; *Jachs et al., 2015*; *Ramsøy and Overgaard, 2004*). With such methods, one could argue that reports of 'unawareness' may only mean some stimuli are *relatively* hard to perceive compared to those that are clearly visible.

To avoid this criterion problem, several groups (*Kolb and Braun, 1995*; *Kunimoto et al., 2001*) sought to identify conditions in which confidence was uncorrelated with accuracy, which they argued would indicate no subjective awareness of the target. Unfortunately, some of these efforts were not replicable (*Morgan et al., 1997*; *Robichaud and Stelmach, 2003*). Others revealed that estimating the correspondence between confidence and accuracy requires mathematical considerations more complicated than originally envisaged (*Evans and Azzopardi, 2007*; *Galvin et al., 2003*; *Maniscalco and Lau, 2012*). Importantly, the conceptual link between metacognitive sensitivity (i.e., correlation between confidence and accuracy) and conscious awareness is itself controversial (*Charles et al., 2013*; *Fleming and Lau, 2014*; *Jachs et al., 2015*).

Here, we employ a recently-developed confidence-rating method to address this problem (*Barthelmé et al., 2009*; *de Gardelle and Mamassian, 2014*). Subjects discriminated two stimulus intervals, only one of which contained a target, and indicated confidence in their decisions using a 2-interval forced-choice procedure (2IFC), that is, indicating which of the two discrimination decisions

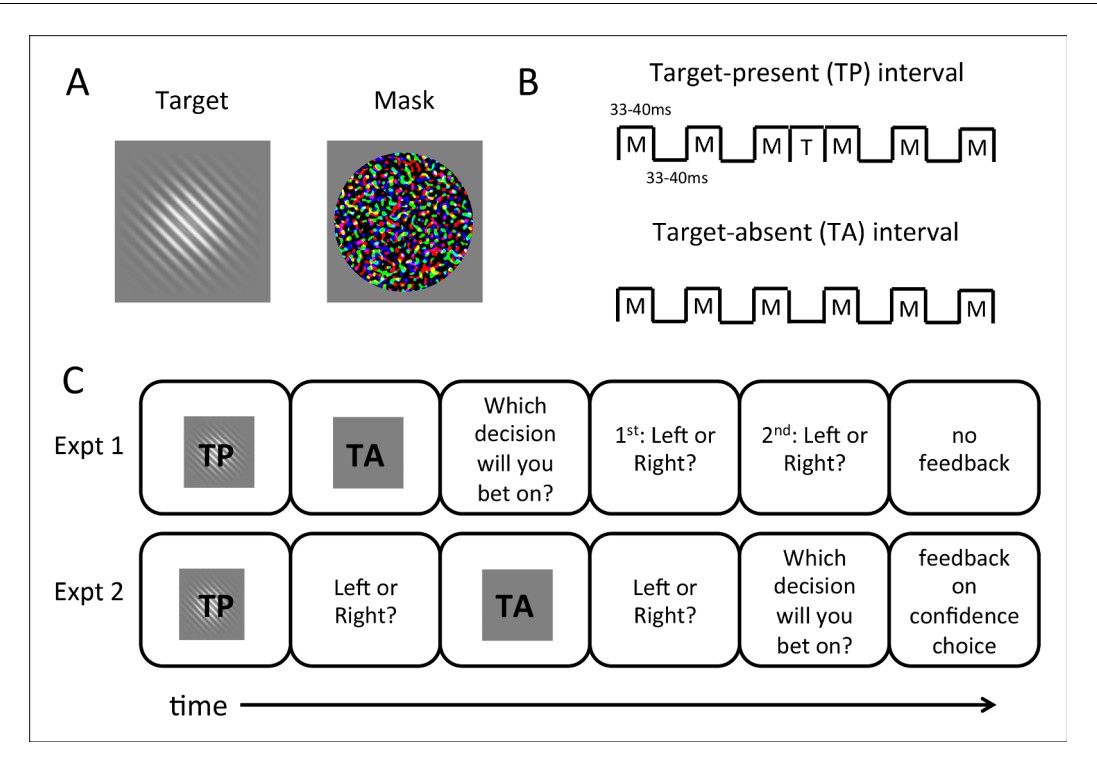

**Figure 1.** Stimuli and procedures for the 2IFC confidence-rating task. (**A**) Targets consisted of oriented (45° left- or right-tilted from vertical) Gabor patches presented at multiple near-threshold contrast levels; masks consisted of bandpass-noise filtered random RGB values (see Materials and methods). (**B**) Each trial consists of two intervals of discrimination in which the target stimulus (**T**) was forward- and backward-masked (**M**). Gabor patch targets were presented only in target-present (TP) intervals; in target-absent (TA) intervals, the target was replaced with blank frames. Otherwise timings of stimuli were matched between the two intervals. (**C**) Experimental tasks. Experiment 1 required subjects to bet on which discrimination they felt more confident before they indicated their orientation discrimination choices (left or right tilt of the Gabor) sequentially for both intervals. Shown is an example trial in which TP is presented before TA; in the experiment this order varied randomly from trial to trial. In Experiment 2, subjects bet on the more confident interval after the discriminations, and feedback was given. (See Materials and methods for more details.)

they felt more confident in. This approach has several advantages. First, 2IFC tasks depend little on response bias compared to multi-point confidence-rating scales. Maintaining the criteria for extensive confidence scales may also be demanding, leading subjects to respond somewhat randomly in conditions of vague awareness and thereby producing the negative result *Kolb and Braun (1995)* observed (*Morgan et al., 1997*). Second, the interpretation of 2IFC confidence-rating in this context is straightforward: 'Performance without Awareness' would mean subjects can perform the target discrimination yet fail to place bets appropriately to distinguish this performance from discrimination of a blank stimulus (which guarantees chance performance). That is, following psychophysics traditions (*Kolb and Braun, 1995*; *Peirce and Jastrow, 1884*), if a certain above-chance discrimination seems introspectively no different from a random guess based on no stimulus at all (as reflected by betting behavior), we interpret the discrimination to be unconscious. Here, we explored whether such Performance without Awareness occurs in normal observers in two behavioral experiments, and compared these results to predictions of a Bayesian ideal observer.

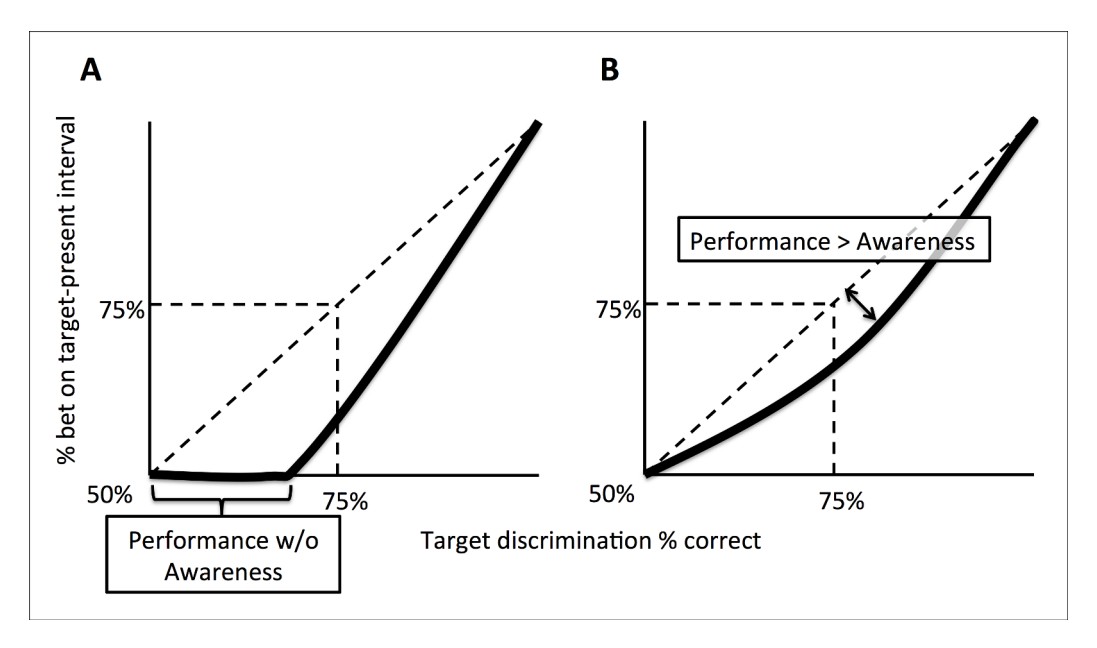

**Figure 2.** Schematic explanation of predictions of the experiments. (**A**) A 'Performance *without* Awareness' pattern of behavior, in which subjects are able to discriminate the target above chance while betting on the target-present interval at chance. (**B**) A 'Performance > Awareness' pattern of behavior, in which subjects are less able to bet on their discrimination decisions than they are able to correctly discriminate the target. In both (**A**) and (**B**), the diagonal dashed line indicates where rate of betting on the target-present interval equals objective discrimination performance.

## Results

### Behavioral experiments

Nine human observers participated in two experiments of our 2IFC confidence-rating paradigm (*Figure 1*). In both experiments, participants viewed two intervals in which they were required to discriminate the orientation (right or left tilt) of a Gabor patch target embedded in forward- and backward-masks (*Figure 1A,B*), and judged which of the discrimination choices they felt more confident in. Crucially, in one of the intervals the target was absent (*Figure 1B*), such that above-chance discrimination performance was impossible. We performed two experiments to assess the potential contributions of question order, receipt of feedback, and a priori knowledge of the presence of a target-absent interval (*Figure 1C*). In Experiment 1, participants judged which decision they felt more confident in and then indicated their orientation decisions for both intervals, while in Experiment 2 they indicated their orientation discrimination decisions before selecting the more-confident interval. In Experiment 2, we also provided feedback on the confidence decision, and told participants that one interval contained no target; this information was withheld from participants in Experiment 1. Stimuli, timing details, and order of question prompts in the two experiments are also discussed in greater detail in the Methods section.

For both experiments, we evaluated whether participants exhibited Performance without Awareness (*Figure 2A*) or Performance > Awareness (*Figure 2B*). In both cases, the response pattern of interest can be visualized as percent of time betting on the target-present interval as a function of percent correct orientation discrimination in the target-present interval. 'Performance without Awareness' (*Figure 2A*) would be supported if observers can discriminate the target above chance (>50% accuracy) while being unable to bet on their choices more often than betting on the target-absent interval (which necessarily yields chance-level performance). That is, observers correctly discriminate the target's orientation more than 50% of the time, but bet on the target-present interval 50% of the time (i.e., they bet randomly on the target-present versus target-absent interval),

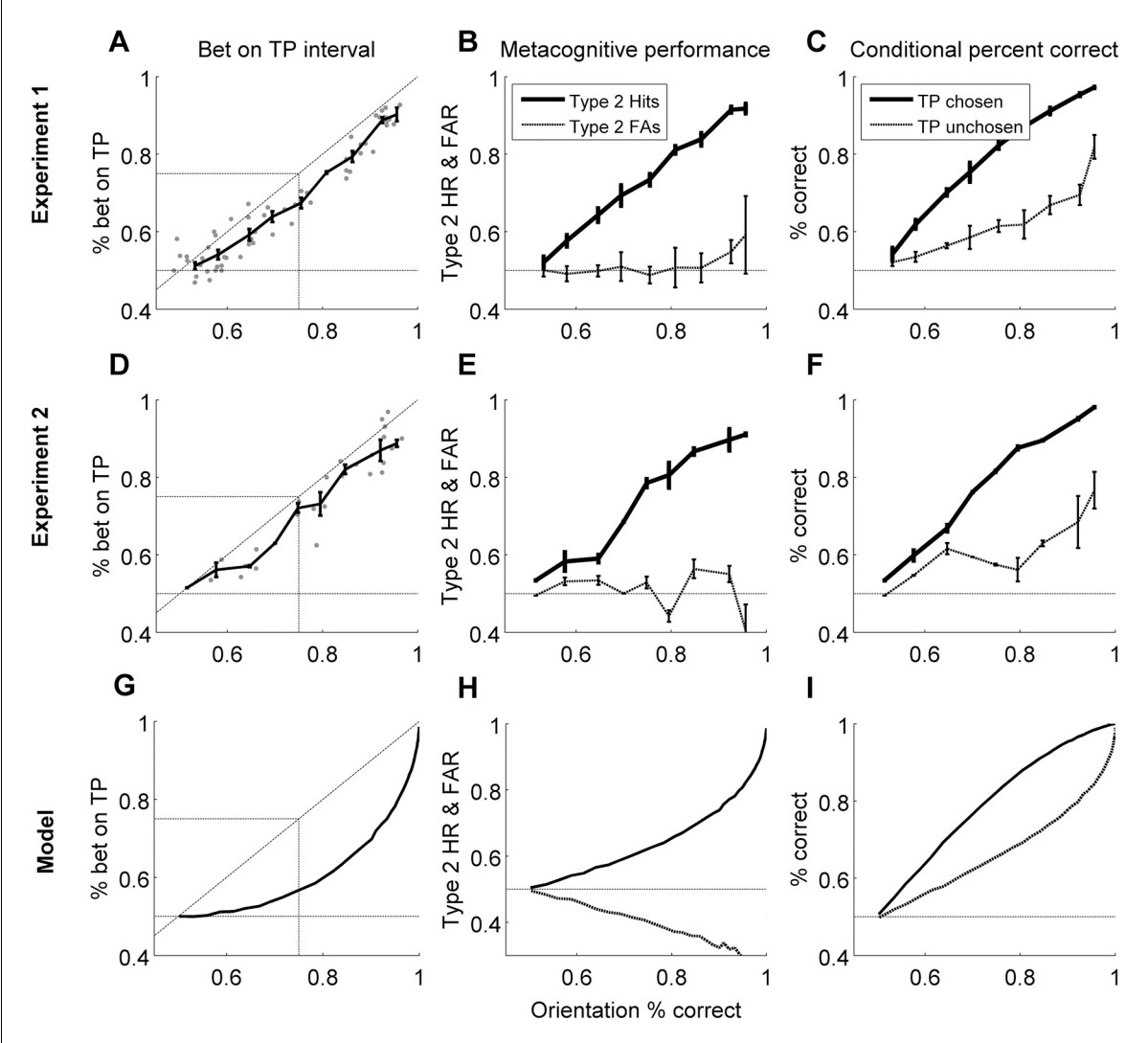

**Figure 3.** Group-level results of behavioral experiments (rows 1 and 2), presented in comparison to the predictions of the Bayesian ideal observer model (row 3; see Materials and methods - Computational Model). In both experiments, human observers displayed no evidence of Performance without Awareness, but appeared to demonstrate Performance > Awareness (panels A and D). However, the ideal observer model also demonstrated such behavior (panel G), indicating that it is not suboptimal at all but arises from the 2IFC nature of the confidence task (see Bayesian Ideal Observer Model results section and *Figure 2* caption for explanation). Horizontal gray lines in panels A, D, and G indicate chance-level betting (50%) on the target-present (TP) interval. Panels B, E, and H show rising Type 2 hit rate ('HR'; when subjects bet on a correct orientation discrimination choice) but relatively flat Type 2 false alarm rate ('FAR'; when subjects bet on an *in*correct orientation discrimination choice), and panels C, F, and I show higher orientation discrimination accuracy when the target-present (TP) interval is bet on; these patterns suggest that human subjects and the Bayesian ideal observer were rating confidence via assessing their probability of correctly discriminating orientation, rather than target presence versus absence only. The model demonstrates good explanatory power for the data across all participants (mean proportion of variance accounted for by the model, $R^2$ = 0.565). Error bars for behavioral data indicate the standard error of the mean across subjects with data in each bin.

indicating they are not aware of the information that contributed to their discrimination decision. If this were to occur, it would most likely happen at low discrimination performance levels, yielding a pattern of behavior similar to that presented in *Figure 2A*.

However, in psychophysics, thresholds can also be defined as midway between ceiling and floor performance (*Macmillan and Creelman, 2004*), such that threshold discrimination performance is defined as 75% accuracy rather than >50% (chance level). This concept can also be applied to sub-jective betting data in the sense that betting on the target-present interval could be considered 'cor-rect' or 'advantageous' betting. In this sense (threshold = 75% correct performance), the subjective

**Table 1.** Individual values, means, standard deviations, and p-values for t-tests showing that Performance > Awareness occurs across both experiments. Results from Experiment 2 show that the pattern does not change with different question order or feedback.

| Expt | Subject | | p(choose TP interval) at p(correct) = 0.75 |
|---|---|---|---|
| 1 | 1 | AVT | 0.676 |
| | 2 | AM | 0.714 |
| | 3 | JDK | 0.716 |
| | 4 | SH | 0.682 |
| | 5 | MM | 0.684 |
| | 6 | AC | 0.685 |
| | 7 | MR | 0.674 |
| | 8 | MK | 0.658 |
| | 9 | RA | 0.619 |
| 2 | 1 | AVT | 0.666 |
| | 2 | AM | 0.713 |
| | 3 | JDK | 0.746 |
| Mean (σ) | | | 0.686 (0.033) |
| t(11) | | | 6.718 |
| p | | | 0.00003 |

threshold for confidence might be above the objective threshold for discrimination. In other words, observers may bet on the target-present interval *less often* than they get the discrimination correct, but still *above chance*. This would occur because the orientation discrimination choice requires evaluation of only one interval (the one with the target in it) and therefore is subject to only one source of uncertainty, but the 'betting' choice requires evaluation of both intervals, and therefore has two potential sources of uncertainty. This pattern of behavior (*Figure 2B*) may occur even if subjects do not display Performance without Awareness, and would be characterized by a pattern of responses that fall below the identity line (diagonal dashed line). We call this possibility 'Performance > Awareness'.

We discuss the results of both experiments together for ease of interpretation, and because the results are very similar (*Figure 3A–F*). To anticipate, we found no evidence of Performance without Awareness. Although we found strong evidence of Performance > Awareness across the experiments (*Figure 3A,D*), subsequent computational modeling (Bayesian Ideal Observer Model section) suggests that this is somewhat trivial: even an ideal observer is expected to show Performance > Awareness (*Figure 3G*; see Bayesian Ideal Observer Model section for further explanation).

To look for evidence of Performance without Awareness, we first plotted percent of trials in which observers bet on the target-present interval against orientation discrimination accuracy for both experiments (*Figure 3A,D*). In contrast to what might have been suggested based on previous results (e.g., *Boyer et al., 2005*; *Charles et al., 2013*; *Merikle et al., 2001*; but see *Snodgrass et al., 2004*), visual inspection alone clearly reveals no evidence for Performance without Awareness in either experiment: it looks as though observers could bet on the target-present interval above chance as soon as they were able to discriminate the target above chance, and there is no hint of the Performance without Awareness pattern. We quantitatively assessed the possibility of Performance without Awareness using a Bayesian observer model (see Modeling Results, below), but found no evidence that a Performance without Awareness pattern could capture human behavior. Individual subjects' performance closely resembles group data and averages (*Appendix 2*).

Because thresholds can be defined in psychophysical terms (75% performance) rather than absolute terms (>50%), we also evaluated the possibility of Performance > Awareness. We used kernel smoothing regression (see Materials and methods) to interpolate each individual subject's data in order to estimate how often subjects bet on the target-present interval when they were performing at 75% correct on orientation discrimination. Because results are very similar across the two

experiments, we combined results from both and performed a two-tailed one-sample t-test to assess whether this predicted percentage betting on the target-present interval significantly diverged from 75%. This analysis revealed that observers bet on the target-present interval significantly less than 75% of the time at 75% correct orientation discrimination accuracy (*Figure 3A,D*, *Table 1*). Thus, observers exhibited Performance > Awareness (but see also Modeling Results, below).

## 2IFC detection?

One possible concern is that subjects were not rating confidence but instead engaging in 2IFC *detection* of the target-present interval. To confirm that subjects were indeed rating confidence, we plotted Type 2 hit rate and Type 2 false alarm rate against orientation discrimination accuracy (*Figure 3B,E*). A Type 2 hit is defined as placing a bet on a correct orientation discrimination decision, whereas a Type 2 false alarm is defined as placing a bet on an *in*correct orientation discrimination decision. These are in contrast to Type 1 hits and false alarms, which can be defined as saying 'left' when a left-tilted Gabor was presented and saying 'left' when a right-tilted Gabor was presented, respectively, according to standard signal detection theoretic definitions (*Green and Swets, 1966*; *Macmillan and Creelman, 2004*).

Subjects displayed increasing Type 2 hit rate as a function of orientation discrimination accuracy, whereas Type 2 false alarm rate remained relatively flat at around 50% (chance level) across increasing orientation discrimination accuracy. In other words, subjects did not bet on orientation discrimination choices they expected to get wrong, even at high performance (i.e. high contrast) levels. Thus, they were probably truly rating confidence and not simply engaging in 2IFC detection. In keeping with this observation, we also plotted orientation discrimination accuracy conditional upon subjects' selection of the target-present interval, i.e. $p(correct_{orientationDiscrimination} \mid target\text{-}present\ selected)$ and $p(correct_{orientationDiscrimination} \mid target\text{-}present\ not\ selected)$ (*Figure 3C,F*). This visualization revealed that subjects were worse at orientation discrimination when they did not select the target-present interval. This result is in keeping with typical observations of worse objective performance for low confidence trials, since not betting on the target-present interval is essentially an indication of low confidence in that discrimination choice. See also the 'Unconscious 'hunches'?' section, below.

Notably, the similarity in participants' behavior between Experiments 1 and 2 reveals that receipt of feedback on confidence judgments, knowledge that one interval is physically blank, question order, and ability to monitor reaction time do not affect behavioral outcomes.

## Unconscious 'hunches'?

Throughout this report, we define conscious awareness of the target to occur when introspective assessment of the correctness of an orientation discrimination choice can differentiate between a target being present or not. In this sense, observers are unconscious of the information contributing to their decision if they can discriminate a target above chance, but doing so feels no different introspectively from discriminating (or guessing about) nothing at all. However, one concern might be that subjects are able to meaningfully rate confidence despite no subjective visual experience of the stimulus due to some sort of non-visual 'hunch' or 'feeling'. Indeed, such metacognitive insights (the ability to introspectively distinguish between correct and incorrect responses) have recently been reported even in the absence of objective task performance sensitivity, although not in the context of perception (e.g., *Scott et al., 2014*).

We think this issue is essentially one of terminology; our definition of conscious awareness follows a long history in psychology and psychophysics traditions in relating the ability to meaningfully rate confidence to subjective awareness (c.f. *Kolb and Braun, 1995*; *Peirce and Jastrow, 1884*), according to which, strictly speaking, a non-visual hunch is also defined as conscious so long as it meaningfully tracks visual processes; regardless of whether such 'hunches' are visual in nature, it is still meaningful to distinguish between having such introspective insight versus having no insight whatsoever. However, we also ran a control study in which the subjective task was to indicate which interval appeared more *visible* rather than confidence in the corresponding discrimination. In other words, it was akin to a 2IFC detection task rather than a metacognitive judgment. Results of this control study (*Appendix 3*) mirrored those of the main experiments: as soon as participants were able to discriminate the target above chance, they were able to indicate which interval contained the target above

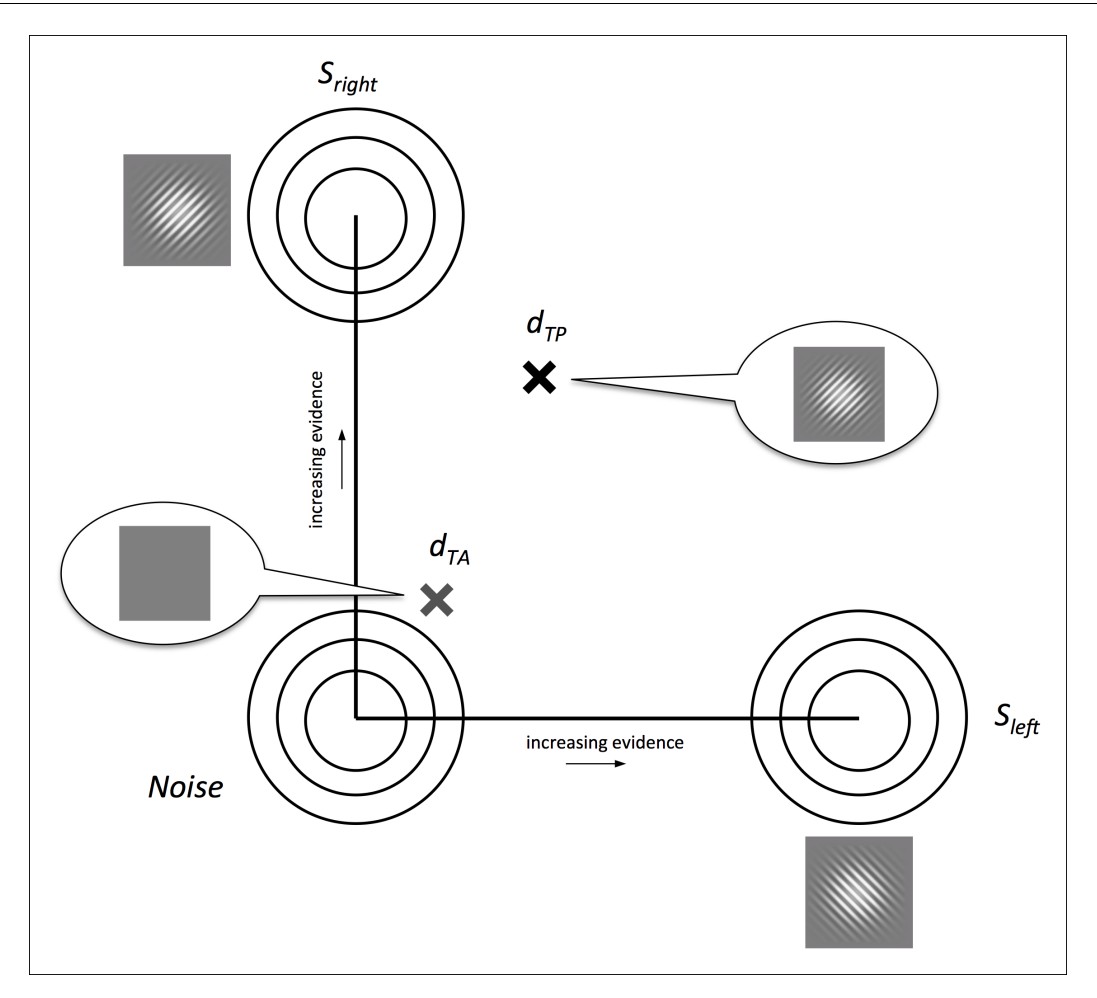

**Figure 4.** Illustration of the Bayesian ideal observer's 2-dimensional representation space, following standard 2-dimensional signal detection theory (*King and Dehaene, 2014*; *Macmillan and Creelman, 2004*). (a) Distributions $S_{left}$ and $S_{right}$ lie on orthogonal axes $c_{left}$ and $c_{right}$ representing left- and right-tilted targets, respectively, and the noise distribution lies at the origin. On each simulated trial, the model 'sees' two samples, one drawn from a source distribution $S_i$ to represent the target-present interval ($d_{TP}$) and the other from the noise distribution to represent the target-absent interval ($d_{TA}$). It marginalizes across all contrast evidence levels to guess the orientations of both samples according to the posterior probabilities of left- and right-tilted sources. Then, it compares the posterior probabilities of the chosen orientations in each interval to select the interval with higher confidence (*p(correct)*) (see Materials and methods - Bayesian ideal observer model).

chance. Thus, even when the 2IFC task was visibility judgment rather than confidence, subjects' behavior was inconsistent with the Performance without Awareness pattern – suggesting there is also no Performance without *Visual* Awareness. See *Appendix 3* for details of the control study.

## Bayesian ideal observer model

We developed a Bayesian ideal observer model utilizing a similar representation space as standard 2-dimensional signal detection theory (*Figure 4*) (*King and Dehaene, 2014*; *Macmillan and Creelman, 2004*). The primary finding is that even an ideal observer model exhibits Performance > Awareness, as depicted in *Figure 1B*. Intuitively, this effect occurs because the orientation discrimination choice requires evaluation of only one interval (the one with the target in it) and therefore is corrupted by only one source of noise, but the 'betting" choice requires evaluation of both intervals, and therefore has two potential sources of noise.

**Table 2.** $R^2$ values quantifying goodness of fit for ideal observer ($\sigma_d = 0$) and three alternative decisional noise magnitudes ($\sigma_d > 0$) which cause increasing degrees of Performance without Awareness. Decisional noise greater than 0 – i.e., increased level of Performance without Awareness – causes a drop in goodness of fit between model and human data. See Methods and **Appendix 4** for more details.

| Expt | Subject | Decisional noise $\sigma_d$ | | | |
| --- | --- | --- | --- | --- | --- |
| | | 0 (Ideal observer) | 0.1 | 0.2 | 0.3 |
| 1 | 1 | 0.465 | 0.459 | 0.456 | 0.447 |
| | 2 | 0.580 | 0.578 | 0.565 | 0.544 |
| | 3 | 0.470 | 0.464 | 0.448 | 0.428 |
| | 4 | 0.396 | 0.392 | 0.381 | 0.363 |
| | 5 | 0.649 | 0.655 | 0.645 | 0.628 |
| | 6 | 0.480 | 0.473 | 0.458 | 0.434 |
| | 7 | 0.453 | 0.452 | 0.444 | 0.427 |
| | 8 | 0.602 | 0.595 | 0.583 | 0.563 |
| | 9 | 0.503 | 0.509 | 0.512 | 0.504 |
| 2 | 1 | 0.624 | 0.624 | 0.622 | 0.612 |
| | 2 | 0.783 | 0.780 | 0.775 | 0.766 |
| | 3 | 0.777 | 0.778 | 0.767 | 0.753 |
| Mean $R^2$ ($\sigma$) | | 0.565 (0.126) | 0.563 (0.128) | 0.555 (0.129) | 0.539 (0.131) |

The model 'performs' a 2IFC confidence discrimination by comparing the posterior probability of left- or right-tilted source distributions given the data to perform the orientation discrimination task on each of the two intervals on each trial. Then, it uses the posterior probability of the choice it made on each interval as a measure of confidence (i.e., *p(correct)*), and compares this measure between the two intervals to select the choice it is more confident in (see **Figure 4** and Materials and methods – Bayesian ideal observer model). We also explored several model variants to establish the robustness of the model's performance; see **Appendix 4** for details on model variants.

Unsurprisingly, the Bayesian ideal observer did not display signs of Performance without Awareness. We next evaluated whether causing the model to exhibit Performance without Awareness (**Figure 2A**) by degrading the 2IFC confidence judgment could produce better fit to participants' data. We tested three levels of increasing decisional noise ($\sigma_d$; see Materials and methods) to cause the model to exhibit increasing Performance without Awareness as described in **Figure 2A**, and assessed the goodness of fit ($R^2$) for each subject for each decisional noise value. We found that causing the model to exhibit increasing Performance without Awareness behavior resulted in increasingly worse $R^2$ values (**Table 2**). To confirm this trend, we conducted a 12 (subjects; subjects 1–3 who completed both experiments are treated independently) x 4 (decisional noise magnitude) repeated measures ANOVA on the $R^2$ values. This analysis revealed a main effect of decisional noise ($F(3,33) = 19.301$, p <0.001), indicating that the ideal observer model ($\sigma_d = 0$) best captures human performance, and that any suboptimal Performance without Awareness ($\sigma_d > 0$) pattern fits human data more poorly than the ideal observer behavior – even without punishing the decisional noise model for having an additional parameter.

Crucially, however, the ideal observer *does* exhibit Performance > Awareness (**Figure 3G**), and to a similar extent as our human participants ($R^2 = 0.565$; see **Appendix 4** for details of goodness of fit metrics); trends for Type 2 hit and false alarm rates (**Figure 3H**), and percent correct conditional upon having bet on the target-present versus target-absent interval (**Figure 3I**), also match human data. That the ideal observer exhibits behavior that may *seem* suboptimal, and in the same pattern as human observers, confirms that this perhaps counterintuitive but optimal behavior arises from the confidence-comparison nature of the 2IFC confidence-rating task: the decision about orientation in

the target-present interval is limited by one source of noise (the single target-present interval), but the comparison of confidence is limited by the system's noise in *both* intervals. So even if confidence monotonically increases with accuracy for the target-present interval, there will be trials in which – by chance – the discrimination choice for the blank (target-absent) interval happens to seem more confident, that is, its posterior probability is larger. This will happen sometimes even on trials in which the observer gets the target-present orientation discrimination correct. In these trials, the observer (human or simulated) will select the target-absent interval. This process will lead to the appearance of what we called Performance > Awareness, as displayed by our human participants and ideal observer (refer also to *Figure 2* for additional explanation). Thus, the subjective ratings by human participants are already close to ideal, as if the actual effective threshold for subjective awareness is no different from the objective threshold for discrimination. Importantly, this is true despite the apparent measured differences in psychophysically defined thresholds (75%).

## Discussion

Blindsight (*Weiskrantz, 1986*) is the intriguing demonstration of Performance without Awareness in neurological patients. Despite widely held beliefs by experts, here we found no evidence that it occurs in normal observers. Importantly, although the measured psychophysical threshold (75%) for awareness seemed to be above the objective discrimination threshold, computational analysis revealed that the *actual effective thresholds* are essentially the same; people's subjective ratings are close to ideal, given their objective performance levels. This challenges longstanding beliefs regarding the nature of subjective versus objective thresholds in perceptual studies (*Merikle et al., 2001*; see survey results in *Appendix 1*).

Our findings cannot rule out all forms of unconscious perception, such as subliminal priming, in which the evidence for unconscious processing is typically indirect benefits in reaction times (*Hannula et al., 2005*). However, our findings bear upon those studies, too. Traditionally, interpreting such effects as unconscious required that the relevant stimuli yield zero sensitivity in a direct task (d' = 0). Recently, many have relaxed this requirement and considered *subjectively* reported lack of awareness as sufficient (*Pessiglione et al., 2009*; *Soto et al., 2011*), presumably because we (wrongly) believed that certain stimuli might surpass the objective threshold while still being below the subjective one. One may also argue that while objective threshold requirements are rigorous, the valid and meaningful measure is the subjective threshold (*Charles et al., 2013*; *Merikle et al., 2001*). Our results suggest this reasoning is flawed. If a stimulus surpasses the objective threshold, there is likely conscious experience; subjects likely *report* lack of awareness because they interpret the response options in relative terms in the context of stimuli of various strengths. This undermines claims that higher-cognitive phenomena – e.g. working memory, error detection, or motivation – can really operate unconsciously, if assessed with reference to subjective rather than objective thresholds (*Charles et al., 2013*; *Pessiglione et al., 2009*; *Soto et al., 2011*).

Although the 2IFC confidence-rating procedure bypasses the response bias problem, interpreting the subjective vs. objective function is non-trivial: to determine whether participants' Performance > Awareness behavior was optimal required detailed computational analysis. An alternative approach, which may be simpler, would be to compare the objective and subjective functions between task conditions, in a rationale similar to *Lau and Passingham (2006)*.

Although we found no evidence of 'blindsight' in normal observers, our study lays out the logic of what would be required to demonstrate it unequivocally. For example, it has recently been argued that TMS-induced 'blindsight' (*Boyer et al., 2005*) is contaminated by criterion bias (*Lloyd et al., 2013*). 2IFC confidence-rating may help resolve such issues without invoking theoretically complicated problems concerning signal detection theory (e.g., *Heeks and Azzopardi, 2015*). Thus, despite their negative nature, our findings may beget fruitful lines of inquiry to address which stimuli, procedures, or brain stimulation techniques can *selectively* impair subjective conscious experience, beyond impacting sheer objective processing sensitivity.

## Materials and methods

### Behavioral experiments

#### Subjects

Twelve subjects (two women, ages 19–32, ten right-handed) gave written informed consent to participate in our behavioral experiments. All subjects had normal or corrected-to-normal eyesight, and wore the same corrective lenses for all sessions, if applicable. Behavioral experiments were conducted in accordance with the Declaration of Helsinki and were approved by the UCLA Institutional Review Board.

#### Stimuli and apparatus

Targets consisted of Gabor patches (sinusoidal gratings) at a spatial frequency of 0.025 cycles/pixel, tilted by 45° to the right or the left of vertical. Gratings and subtended 500 pixels, or ~111 visual degrees, and were presented in a circular annulus with a Gaussian hull spatial constant of 100. On each trial, targets could take on one of thirteen possible contrast levels drawn from the range 15–90%. Masks consisted of white noise patches of random RGB values bandpass-filtered to a range of spatial frequencies immediately surrounding the spatial frequency of the target. They were presented in a circular annulus of identical size to the spatial envelope of the Gabor patch targets. All stimuli were displayed via a custom Matlab R2013a (Natuck, MA) script utilizing PsychToolbox 3.0.12 on a gamma-corrected Dell E773c CRT monitor with a refresh rate of 75 Hz.

#### Procedure – Experiment 1

Nine subjects participated in Experiment 1. Subjects were seated with their chins in a chinrest at a viewing distance of 42 cm from the screen. Targets and masks (*Figure 1A*) were presented for two to three frames (33–40 ms) each (jittered timing, with equal probability for two or three frames), with 33-40 ms ISI between masks and 0ms ISI for target-mask or mask-target transitions, in a forward- and backward-masking paradigm in which three masks were presented before and three after the target presentation (i.e., the target was 'sandwiched' between mask presentations) (*Figure 1B*).

The trial structure extends the two-by-two forced-choice (2x2FC) paradigm first introduced by *Nachmias and Weber (1975)* and subsequently employed to explore the relationship between detection and identification (e.g., *Thomas et al., 1982*; *Watson and Robson, 1981*), and more recently applied to research on confidence (*Barthelmé et al., 2009*, *2010*; *de Gardelle and Mamassian, 2014*). We combined these procedure types. In our procedures, each trial consists of two time intervals, within only one of which the target is presented. In target-absent intervals, the target presentation was replaced with blank frames, similar to the blank frames between masks, to maximize phenomenological similarity between target-present (TP) and target-absent intervals (TA) (*Figure 1B*). Unlike previous usage of the 2x2FC, however, we required observers to indicate target orientation on *both* target-present and target-absent intervals within a trial in addition to the final judgment type, despite the fact that there was a target in only one of the intervals.

In target-present intervals, targets were presented at 45° tilted right or left from vertical at one of the possible contrasts. Following presentation of both intervals, observers pressed a key indicating which discrimination decision they would like to bet on (a measure of confidence; Type 2 judgment), and then indicated their discrimination choices for both intervals in order (leftward or rightward tilt; Type 1 judgment) (*Figure 1C*). In target-absent intervals, participants' answers were coded as 'correct' with 50% probability. No feedback was provided on a trial-by-trial basis. To motivate subjects, we informed them that a target was present in both intervals, but that one might be harder to discriminate than the other. Subjects were informed that they would be awarded a point for every correct discrimination (Type 1 judgment), and an additional point every time they bet on an interval they discriminated correctly (Type 2 judgment), and total points were displayed at the end of the experiment; they were also told that if they earned more points than the previous participant, they would be paid an additional $10 bonus at the end of all sessions.

In each behavioral session, trials were presented in a randomized full factorial design, counterbalancing interval order, in ten blocks of 52 trials per block. Every subject undertook five 60-minute sessions, for a total of 2600 trials spread across up to thirteen contrast levels, two orientations, and two interval presentation orders. Levels of contrast presented to each participant were titrated across

sessions to ensure performance spanning approximately evenly from chance (50% correct) to 100% correct, resulting in no fewer than 200 trials per contrast level (10 trials per condition x 2 orientations x 2 interval orders x 5 sessions). Subjects were paid $10 per session.

## Procedure – Experiment 2

Three subjects who had participated in Experiment 1 also participated in Experiment 2. Procedures for Experiment 2 were identical to those described above for Experiment 1, except for the feedback structure, observer's knowledge about target-present versus target-absent intervals, and order of questions (*Figure 1B*). In Experiment 2, we wanted to motivate subjects to bet on the target-present interval as much as possible, to maximize the possibility of observers performing optimally (i.e., to alleviate any Performance > Awareness). So, we defined a 'correct' Type 2 judgment for the purposes of feedback only as a Type 2 hit, i.e. trials in which the observer correctly discriminated the target-present interval and bet on the target-present interval. Subjects were also informed that in one of the intervals the target was physically absent, and that betting on that interval would not earn them a point even if they 'discriminated' its orientation correctly (as before, they still had a 50% chance to earn a point for 'correctly discriminating' the target-absent interval; subjects were made aware of this structure). Additionally, we provided 'correct/incorrect' feedback on the Type 2 responses to further encourage betting on the target-present interval. Finally, we altered the question order such that after each interval was presented, subjects pressed a button to discriminate the interval, and then only after both intervals had been presented did they indicate which choice they would like to bet on. In this way, subjects were allowed the ability to monitor their own reaction times, which ought to be faster for target-present intervals on average (as target-absent intervals are simply guesses by definition); this would provide another source of potential information to contribute to confidence judgments, as it has been shown that subjects use reaction time monitoring to inform confidence judgments (*Kiani et al., 2014*). Points were awarded as in Experiment 1, and the same bonus payment motivation was employed. Also as before, participants completed five behavioral sessions each for Experiment 2, and were paid $10 per session.

## Statistical analyses

For each subject in each experiment, data were collapsed across tilt (left/right), interval presentation order (first/second), and session for each contrast level. At each contrast level for each subject, we next calculated (a) percent correct orientation discrimination, (b) percent of trials in which the target-present interval was chosen, (c) Type 2 hit rate and Type 2 false alarm rate according to standard Type 2 signal detection theoretic definitions (Type 2 hit: correct orientation discrimination and bet on target-present interval; Type 2 false alarm: incorrect orientation discrimination and bet on target-present interval) (*Fleming and Lau, 2014*; *Maniscalco and Lau, 2012*), and (d) percent correct orientation discrimination conditional on having chosen the target-present versus target-absent interval.

Group-level analyses and graphical presentation were conducted by binning subjects' data into ten equally-spaced bins of percent correct orientation discrimination performance in the range 0.5 – 1 and calculating the mean and standard deviation of each of the above statistics for each bin.

To interpolate between discrete data points, we fitted a kernel smoothing regression function to each observer's data, which is a non-parametric approach to estimate the conditional expectation of a random variable, $E(Y|X) = f(X)$ where $f$ is a non-parametric function. This approach is based on kernel density estimation, implementing Nadaraya-Watson kernel regression (*Nadaraya, 1964*; *Watson, 1964*) via

$$\hat{f}(x; K, h) = \frac{\sum_{i=1}^{n} K_h(x - x_i) y_i}{\sum_{i=1}^{n} K_h(x - x_i)} \tag{1}$$

where $K$ is a Gaussian kernel with bandwidth $h$.

All analyses were carried out in Matlab R2013a (Natuck, MA) and SPSS Version 22 (IBM Corporation; Armonk, NY).

## Bayesian ideal observer model

### Model space

Our model representation space extends *Macmillan and Creelman's (2004)* two-dimensional signal detection theory (SDT) and related Bayesian (*King and Dehaene, 2014*) framework, in which stimulus categories are represented by bivariate Gaussian distributions centered along the axes in a Cartesian plane, and 'noise' (or a blank stimulus) is represented by a similar bivariate Gaussian centered at the origin (*Figure 4*). Although for this particular task we could have used a 1-dimensional space alternative (see e.g. *Sridharan et al., 2014*), to facilitate additional model variants (see *Appendix 4*) and possible future applications to stimuli that contain a mixture of multiple stimulus categories, we elected to present the model in a two-dimensional format. To accomplish both the orientation discrimination and 2IFC confidence judgments, on each simulated two-interval trial, two pairs of evidence values (representing the evidence in favor of a left- or right-tilted target) of the form $d = [d_{left}, d_{right}]$ are drawn: one sample is drawn from one of the signal distributions $S$ ($d_{TP}$, target-present intervals), and the other drawn from the noise distribution ($d_{TA}$, target-absent intervals) (*Figure 4*).

### Inference process

Our ideal observer employs Bayesian inference in which each interval's sample (i.e., evidence pair) $d$ is first categorized as belonging to $S_1$ or $S_2$ on the basis of the posterior probabilities of each, and then uses the posterior probability of the chosen orientation as a measure of confidence in each discrimination decision.

We assume that each generating stimulus category, $S$, is dependent on the evidence in favor (or contrast) of the presented stimulus, $c$, and can be represented by a bivariate Gaussian distribution such that $S_{c_{left}} \sim N([c, 0], \sum)$ for a 'left' tilt and $S_{c_{right}} \sim N([0, c], \sum)$ for a 'right' tilt (*Figure 5*). Additionally, in the most basic formulation we define $\sum = \begin{bmatrix} 1 & 0 \\ 0 & 1 \end{bmatrix}$ (although we explore other potentially more biologically plausible variants; see *Appendix 4*). We also assume the $c_{left}$ and $c_{right}$ axes (left and right tilt) to be orthogonal, although this constraint is not necessary for the model to capture behavioral performance (see *Appendix 4*).

Importantly, $c$ – the contrast or evidence level along each axis that gave rise to the data sample the observer sees – is unknown to the observer. So, because contrast evidence is a secondary (or nuisance) variable to the primary variable of interest – in this case, the orientation of the Gabor patch – the observer 'integrates out' or marginalizes over all possible contrast evidence levels to produce the posterior probability estimate of each tilt (*Yuille and Bülthoff, 1996*). Thus, the joint probability of each orientation and contrast evidence level is estimated through Bayes' rule

$$p(S, c|d) = \frac{p(d|S, c)\, p(S, c)}{p(d)} \tag{2}$$

and then the secondary variable is integrated out, leaving estimation of the posterior probability of each orientation $S$ via the marginal distribution (*Yuille & Bülthoff, 1996*)

$$p(S|d) = \int p(S, c|d) dc \tag{3}$$

In the simplest form, both orientations have equal prior probability of 0.5. The observer then makes its orientation decision (for each interval) via

$$S_{chosen} = \arg\max_i p(S_i|d) \tag{4}$$

To determine which interval's choice the observer is more confident in, the model refers to the magnitude of the posterior probabilities of each $S_{chosen}$ in each interval as a measure of the probability of having made a correct orientation discrimination choice, i.e. *p(correct) = p($S_{chosen}$|d)*. Then, the observer compares these posterior probabilities for the target-present (TP) and target-absent (TA) intervals by computing a decision variable $D$ via

$$D = \log\left(\frac{p(S_{chosen, TP}|d_{TP})}{p(S_{chosen, TA}|d_{TA})}\right) \tag{5}$$

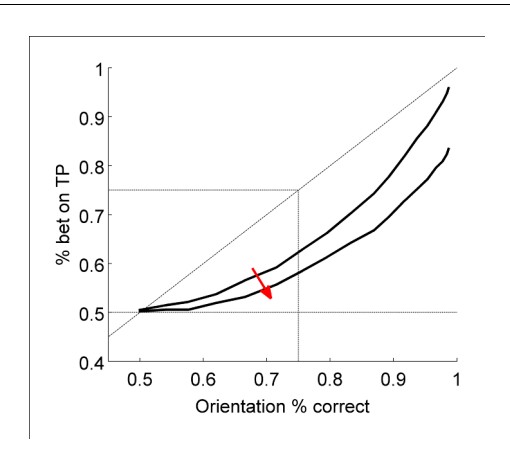

**Figure 5.** Illustration of increasing values for $\sigma_d$ on the appearance of Performance without Awareness behavior, used to evaluate the possibility that human participants may have exhibited Performance without Awareness. Increasing $\sigma_d$ values resulted in increasingly poor $R^2$ values (see Results), indicating that the ideal observer (which displays no performance without awareness) produces the best fit to human data.

The observer bets on the interval with the higher probability of being correct: if this decision variable $D$ is greater than 0, the observer selects the target-present interval to 'bet' on; if it is less than 0, the observer selects the target-absent interval. Sample code for this Bayesian ideal observer is included in *Source code 1.*

## Evaluation of model performance

We examined the relative agreement between our model's predictions and collected behavioral data by calculating the multinomial likelihood of the model given the observed data, which has previously been used within a signal detection framework. Details of goodness of fit calculations are described in *Appendix 4*.

To evaluate whether human participants exhibited Performance without Awareness, we needed to cause the model to also exhibit Performance without Awareness. We therefore degraded the 2IFC confidence judgment process in the following way: On each trial, after the orientation decision had been reached, we programmed an added decisional noise parameter, $\sigma_d$, such that the decision variable D calculated as in Equation 5 was corrupted by additive Gaussian noise with mean 0 and standard deviation $\sigma_D$, such that

$$D = \log\left(\frac{p(S_{chosen,TP}|d_{TP})}{p(S_{chosen,TA}|d_{TA})}\right) + \sigma_d \qquad (6)$$

This causes the model to perform closer to chance at higher levels of orientation discrimination performance, i.e. to exhibit Performance without Awareness at increasing objective performance levels (*Figure 5*). We tested three decisional noise magnitudes – 0.1, 0.2, and 0.3 – and calculated the goodness of fit (see *Appendix 4*) for each $\sigma_d$ for each subject.

## Alternative models

We also examine three other possible contributing factors: correlated noise/non-orthogonal source distributions, signal-dependent (multiplicative), and signal-independent (additive) noise (see *Appendix 4*). These factors do not affect the qualitative trend of the model's performance.

For completeness, we also examine two other decision rules, detailed in *Appendix 5*: a heuristic observer which does not ignore contrast evidence as above, but explicitly estimates the most likely contrast level via hierarchical Bayesian inference (*Yuille and Bülthoff, 1996*); and a heuristic likelihood comparison observer (similar to *Barthelmé et al., 2009*). Importantly, the hierarchical model produced behavior similar to the ideal observer, indicating that such behavior is not idiosyncratic or

specific only to the ideal observer presented above. The likelihood-only model, on the other hand, failed to produce predictions that matched collected behavioral data, either qualitatively or quantitatively.

## Acknowledgements

This work was supported by the National Institute of Health (US) to HL (grant number R01NS088628). We thank Brian Maniscalco, Dobromir Rahnev, Hongjing Lu, and Zili Liu for helpful comments.

## Additional information

### Funding

| Funder | Grant reference number | Author |
| --- | --- | --- |
| National Institutes of Health | R01NS088628 | Hakwan Lau |

The funders had no role in study design, data collection and interpretation, or the decision to submit the work for publication.

### Author contributions

MAKP, Conception and design, Acquisition of data, Analysis and interpretation of data, Drafting or revising the article; HL, Conception and design, Analysis and interpretation of data, Drafting or revising the article

### Ethics

Human subjects: Eleven subjects (two women, ages 19-32, ten right-handed) gave written informed consent to participate in our behavioral experiments. All subjects had normal or corrected-to-normal eyesight, and wore the same corrective lenses for all sessions, if applicable. Behavioral experiments were conducted in accordance with the Declaration of Helsinki and were approved by the UCLA Institutional Review Board. Eighty-seven respondents replied to our informal online survey. Survey procedures were conducted in accordance with the Declaration of Helsinki and were approved by the UCLA Institutional Review Board. Thus, all survey respondents provided informed consent to participate in the informal online survey, and behavioral subjects provided written informed consent to participate in the behavioral experiments.

## Additional files

### Supplementary files

• Source code 1. Ideal observer model.

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

## Appendix 1: Survey

### Methods

#### Subjects

Eighty-seven respondents replied to our informal online survey. Survey procedures were conducted in accordance with the Declaration of Helsinki and were approved by the UCLA Institutional Review Board.

### Survey questions

All questions were presented through the online software SurveyMonkey. The survey was advertised through use of social media (e.g. Facebook, Twitter), and through placement of an advertisement soliciting volunteer participation in the Association for the Scientific Study of Consciousness (ASSC) Monthly Newsletter. Respondents were not paid for their answers. This survey was informal, in that it did not involve random sampling or counterbalancing of question order.

We asked three critical questions of respondents:

1. "Do you personally believe that it is possible for a stimulus to be perceived subliminally, i.e., to exert influence on neural processing and behavior without the relevant feature being perceivable at all?"

2. "Do you personally believe that for some stimulus, there is a subjective threshold (for the subjective, conscious experience of seeing to occur) above an objective threshold (for one to be able to discriminate or identify the stimulus)? That is, do you think there exists a certain contrast level or duration of presentation such that subjects will be able to discriminate or identify the stimulus even though they cannot subjectively see the stimulus?"

3. "If you answered 'yes' to the last question, do you think it has been demonstrated unequivocally in the literature?"

### Results

Eighty-seven individuals responded to our informal survey. We collected demographic data on the following: age, highest degree earned, year of highest degree, field of degree earned, current occupation/position, and number of publications related to subliminal perception. Most respondents held a research doctorate (66%), mostly in Psychology (39%), Philosophy (19%), or Neuroscience (18%). About half of respondents (52%) received their degree in the last five years, and just over a third (38%) reported having one or more relevant publications on subliminal/unconscious perception. See *Appendix 2—Figure 1* for detailed demographic information.

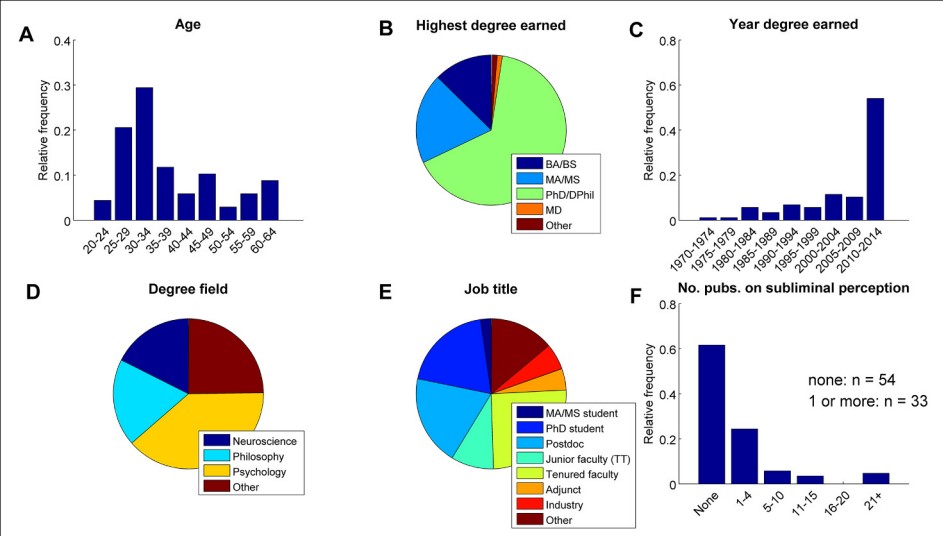

**Appendix 1—Figure 1.** Demographics of survey respondents (n = 87).

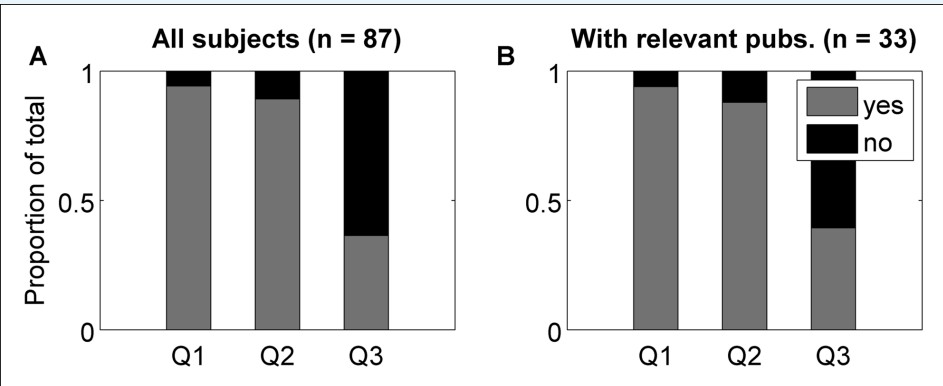

**Appendix 1—Figure 2.** Answers to the three survey questions of interest.
(A) Most respondents reported believing that some forms of subliminal perception exists (Q1), and that specifically the subjective threshold for conscious awareness is above the objective threshold for direct discrimination (Q2). However, most respondents also reported that they did not think this phenomenon (Q2) had been convincingly demonstrated in the literature (Q3). (**B**) The pattern of results was similar for the subset of respondents who had at least one relevant publication.

Most respondents reported believing that subliminal processing exists (Q1) (94%), and that the subjective threshold is above the objective one (Q2) (89%). Of those who responded 'Yes' to Question 2, however, only about a third (36%) reported believing Performance without Awareness has been unequivocally demonstrated in the literature (Q3) (Appendix 2—Figure 2A). Interestingly, the pattern of responses did not change for the subset of respondents who reported having at least one relevant publication (Q1: 94%, Q2: 88%, Q3: 39%; *Appendix 2— Figure 2B*). Thus, although a majority of scholars in this field report believing Performance without Awareness can exist, most also recognize that demonstrating it convincingly has been difficult.

## Appendix 2: Individual behavioral results: Experiments 1 and 2

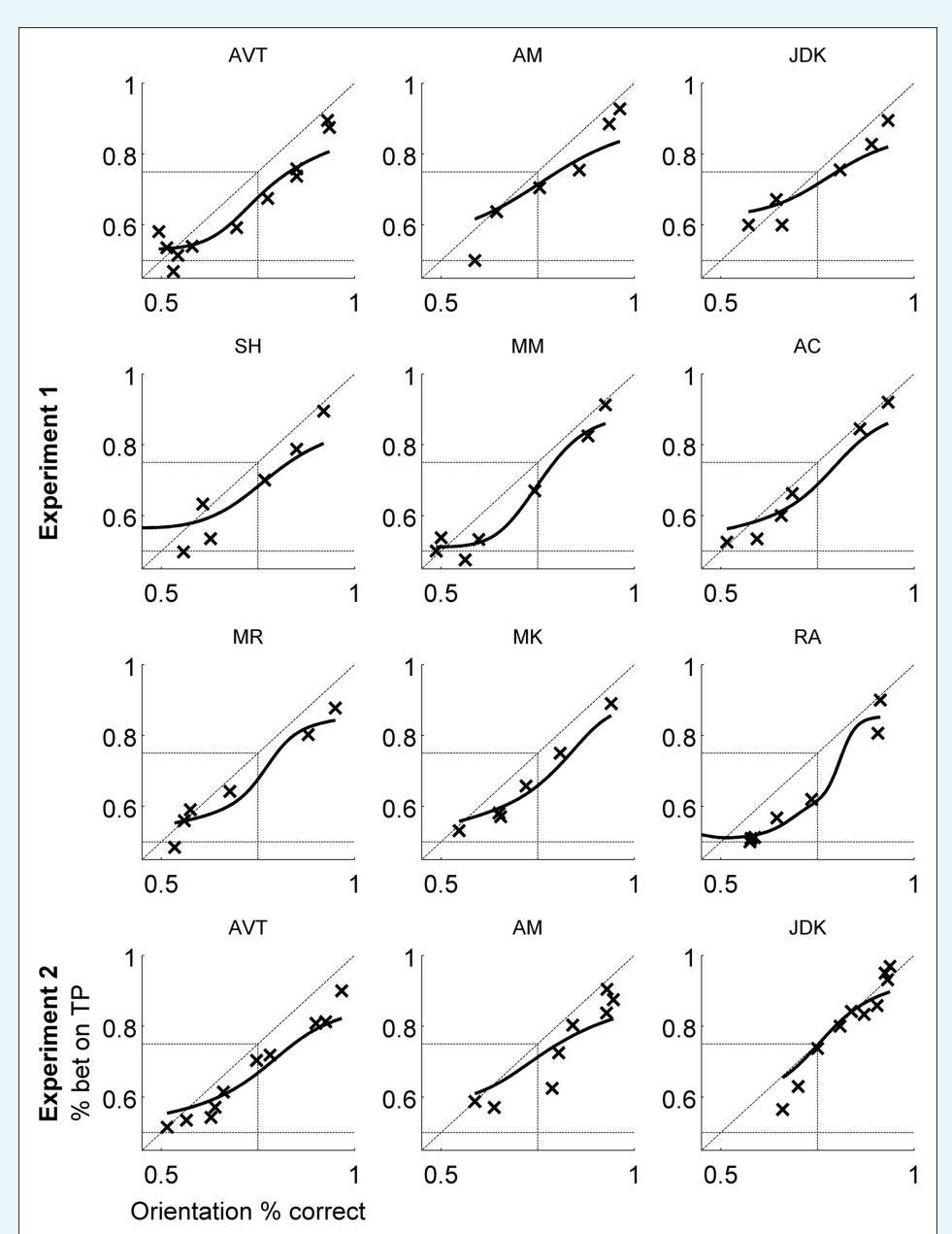

**Appendix 2—Figure 1.** Individual subjects' data for percent betting on target-present (TP) interval as a function of orientation discrimination percent correct.

Individual subjects' data closely resembles grouped data: all subjects exhibited Performance > Awareness, but no subjects exhibited Performance without Awareness. The same three subjects participated in both Experiments 1 and 2.

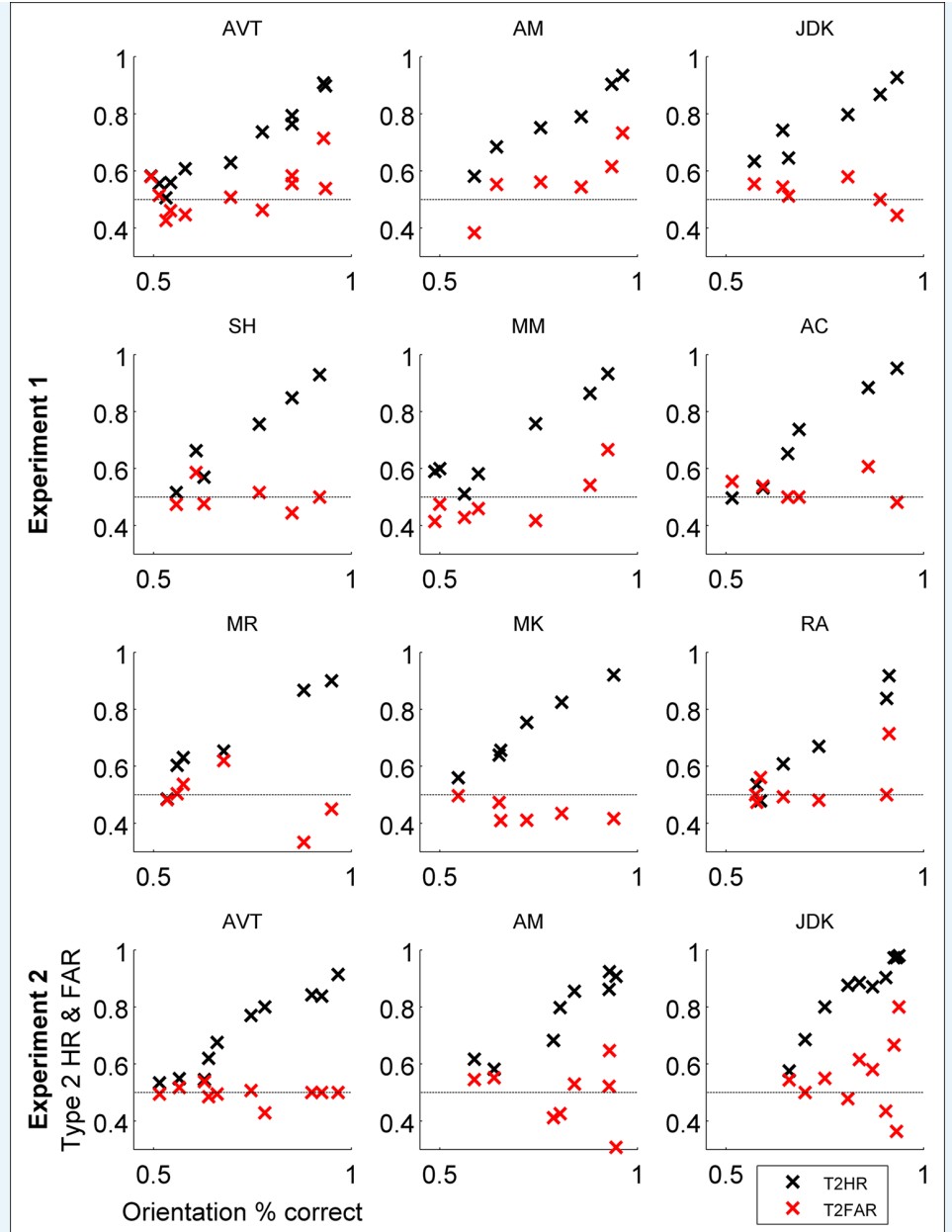

**Appendix 2—Figure 2.** Individual subjects' data for Type 2 hit rate (T2HR) and Type 2 false alarm rate (T2FAR).
Individual subjects' data closely resembles grouped data. The same three subjects participated in both Experiments 1 and 2.

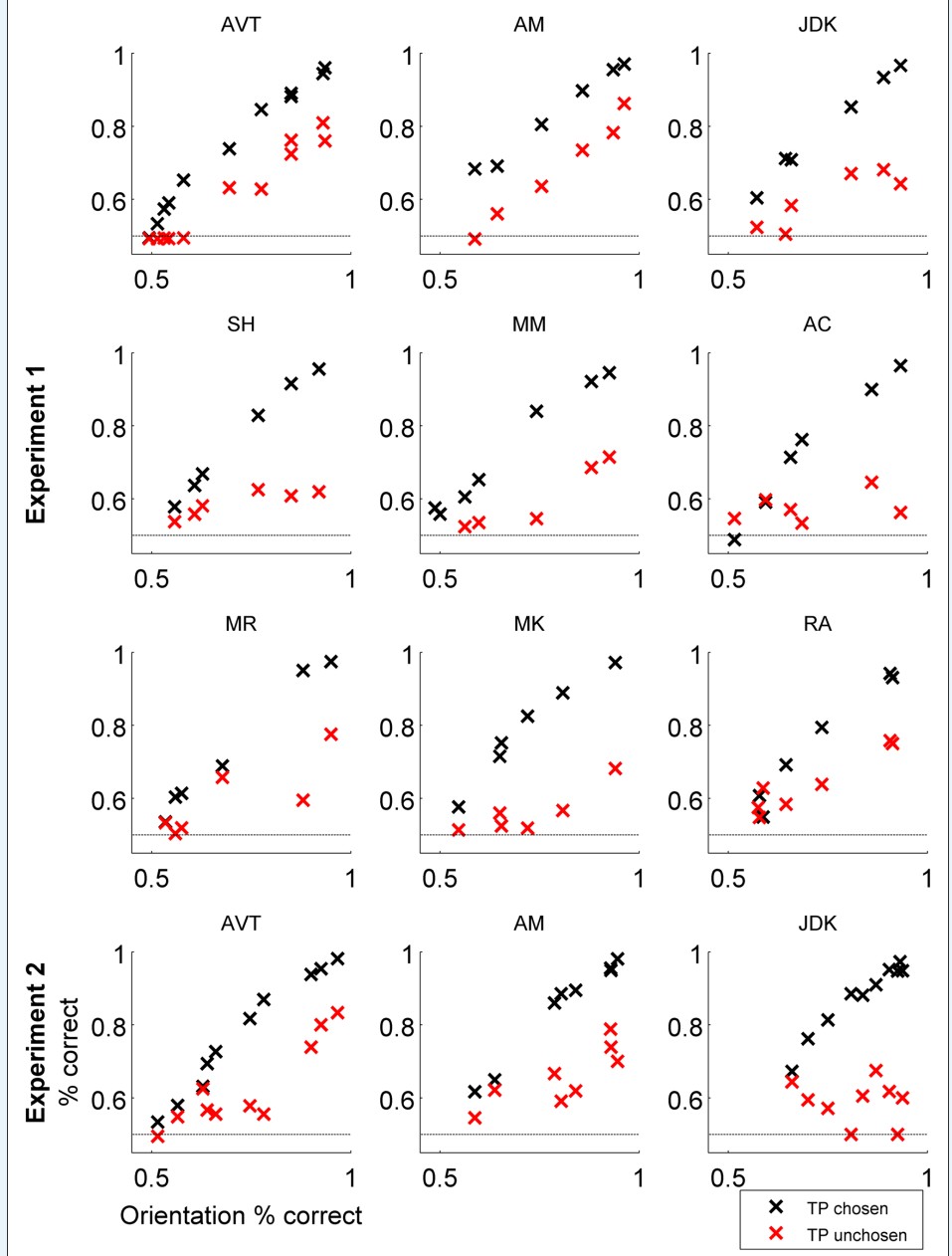

**Appendix 2—Figure 3.** Individual subjects' data for percent correct orientation discrimination conditional on having chosen to bet on the target-present (TP) interval or not. Individual subjects' data closely resembles grouped data.

The same three subjects participated in both Experiments 1 and 2.

We also found that subjects were overall unbiased in their choices of left- versus right-tilted Gabor patches, justifying use of percent correct rather than the signal detection theory metric d'. We calculated the mean criterion used by each subject across all levels of contrast according to signal detection theory, assuming equal variance for distributions representing left- versus right-tilted stimuli, i.e.

$$c = -\frac{1}{2}(z(HR) + z(FAR)) \qquad (A2-1)$$

where HR refers to the hit rate (i.e., the participant said 'left' when a left-tilted Gabor was presented) and FAR refers to the false alarm rate (i.e., the participant said 'left' when a right-tilted Gabor was presented) (*Macmillan and Creelman, 2004*). Individual subjects' criterion values are presented in *Appendix 2–Table 1*. A two-tailed t-test (treating each subject as an independent sample) revealed these values to not be significantly different from 0 (*Appendix 2–Table 1*).

**Appendix 2–Table 1.** Mean value of criterion for each subject, demonstrating that subjects did not display biases to say 'left' versus 'right' in the orientation discrimination task.

| Experiment | Subject | c |
|---|---|---|
| 1 | 1 | 0.047 |
| | 2 | 0.074 |
| | 3 | -0.039 |
| | 4 | -0.029 |
| | 5 | -0.013 |
| | 6 | -0.030 |
| | 7 | 0.0267 |
| | 8 | -0.039 |
| | 9 | 0.003 |
| 2 | 1 | 0.229 |
| | 2 | 0.004 |
| | 3 | 0.035 |
| Mean (σ) | | 0.022 |
| t(11) | | 1.04 |
| p | | 0.3215 |

# Appendix 3: Control study

We ran a control study to assess whether the lack of Performance without Awareness observed in Experiments 1 and 2 may have occurred because subjects have an unconscious 'hunch' of confidence despite having absolutely no phenomenal experience of the stimulus. Three observers (all male, ages 19-32, all right-handed) gave written informed consent to participate. One observer had participated in Experiments 1 and 2; the other two were naive. All procedures were identical to Experiment 2, with two exceptions: (1) observers were not told that one interval was blank; and (2) observers were asked to indicate which interval was 'more visible' rather than which discrimination they believed they were more likely to get correct. Thus, observers engaged in a task more akin to a 2IFC detection task rather than a metacognitive judgment.

Despite these manipulations, the results of the control experiment closely mirror the results of Experiments 1 and 2. Subjects were able to detect the target in the TP interval above chance as soon as they were able to discriminate the target above chance, and individual subjects' data closely resembled the group data (Figure S3). Note that the computational model described in the main text does not apply to these data, as participants were doing 2IFC detectability discrimination rather than a 2IFC metacognitive judgment. This means that even though participants' data lie near the identity line of equal orientation percent correct and percent betting on the TP interval (dashed diagonal line) – *above* the behavior predicted by the confidence model – this should not be taken to mean they are displaying any sort of 'supra-optimal' behavior, as the tasks are fundamentally different. The position of responses on this graph for this task will depend on the precise relationship between the detectability and discriminability of the stimulus for each individual subject (*Thomas et al., 1982*; *Thomas, 1987*).

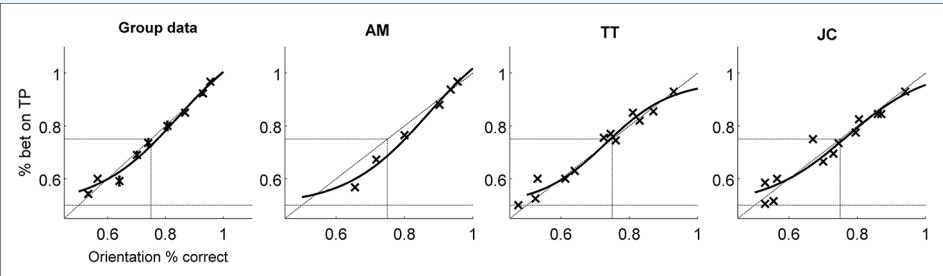

**Appendix 3—Figure 1.** Results of control experiment.
Psychometric functions were fitted as in Experiments 1 and 2 (see Materials and methods — Statistical analyses).Group data is representative of individual subjects' responses (left panel versus right three panels). As before, subjects displayed behavior inconsistent with Performance without Awareness. If anything, participants were able to bet on the target-present interval *more* often when the interval judgment was one of detection rather than Type 2 or metacognitive assessment of the correctness of the Type 1 discrimination. However, this should not be taken to indicate 'supra-optimal' behavior with reference to the ideal observer model of confidence, as the task here is not confidence but a task akin to 2IFC detection, meaning it would require a different computational analysis for assessment of optimality.

# Appendix 4: Model details and variants

## Adjustments for non-orthogonal signal dimensions

In its simplest form, the ideal observer model assumes independence/orthogonality of the stimuli – graphically, that the $S_{left}$ and $S_{right}$ distributions lie on the x and y axes, respectively. However, it is conceivable that the stimuli are correlated or anti-correlated. Therefore, we explored modifying the $c_{right}$ axis to define it as a vector specified by $\theta$, the angle between the x axis and the vector (**Appendix 4—Figure 1**).

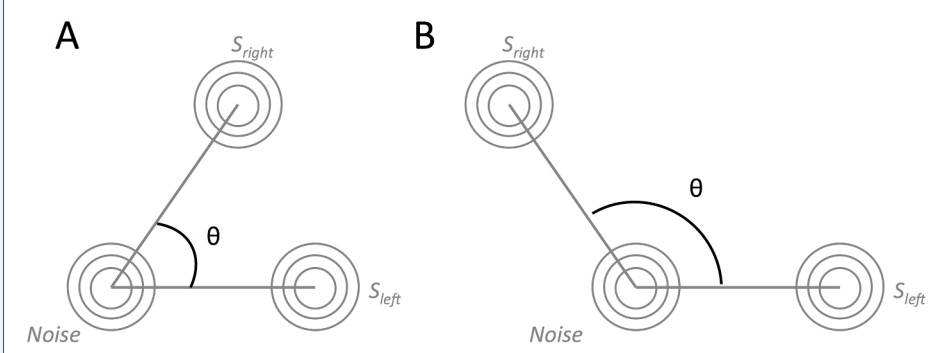

**Appendix 4—Figure 1.** Graphical intuition of correlated stimuli (**A**) and anti-correlated stimuli (**B**).

Formally, instead of $c_{right} = [0, c]$, we define

$$c_{right} = [c \cdot cos(\theta), c \cdot sin(\theta)] \qquad (A4-1)$$

The $\theta$ parameter thus allows control of the degree of correlation (or anti-correlation) between the $S_{left}$ and $S_{right}$ distributions. This formulation is mathematically equivalent to specifying non-zero covariance in the definition of the variance-covariance matrix $\Sigma$ (Macmillan & Creelman, 2004), but we favor using $\theta$ here because it provides a graphically intuitive analog to changing the amount of 'tilt' between the left- and right-tilted Gabor patch targets, that is, the relationship between the detectability and discriminability of a stimulus (see also **King and Dehaene, 2014**). For example, Gabor patches that are ± 45° tilted from vertical ought to demand a larger $\theta$ value than other, less discriminable versions, such as ± 5° tilt.

To explore whether this factor was important to model fits, we fitted the value for $\theta$ to the three participants' data who had completed both main experiments (see Comparison of model to behavioral data, below) and compared model fits with measures of $R^2$ (see Goodness of fit, below) and the Bayesian information criterion (BIC).

Best-fitting values for $\theta$ were quite consistent across participants, and indicate a slight anti-correlation between the $S_{left}$ and $S_{right}$ distributions ($\mu = 97.27°$, $\sigma = 18.79°$). The goodness of fit was evaluated for these three participant's data individually for each experiment given these best-fitting values, and was found to be very good for all across two different null models ($R^2 = 0.636 \pm 0.172$; BIC = $4669.31 \pm 960.94$). However, the single free parameter $\theta$ does not provide any significant improvement in the fit of the model to the data: with no free parameters, $R^2 = 0.617 \pm 0.141$ and BIC = $4657.10 \pm 960.73$. Thus, we present the parameter-free model in the main text.

## Comparison of model to behavioral data

We examined the relative agreement between our model's predictions and collected behavioral data by calculating the multinomial likelihood of the model given the observed data (**Dorfman and Alf, 1969**), which has previously been used within a signal detection framework. Formally, the likelihood of a certain model $m$ (with a given set of parameters $\varphi$) can be expressed as,

$$L_m(\phi|data) \propto \prod_{ij} P_\phi \left(R_i|S_j\right)^{n_{data}(R_i|S_j)}$$

$$(A4-2)$$

where each $R_i$ is a behavioral response a subject may produce on a given trial, and each $S_j$ is a type of stimulus that might be shown on that trial. The expression '$n_{data}(R_i S_j)$' is a count of how many times a subject actually produced $R_i$ after being shown $S_j$. The expression '$P_\phi(R_i S_j)$' denotes the probability with which the subject produces the response $R_i$ after being presented with $S_j$, according to the model specified with parameters $\phi$. This corresponds to the percentage of time each of the models described above produced response $R_i$ after having been 'presented' with stimulus $S_j$. Note that this approach does not examine the performance of a model relative to the behavioral data with reference to any summary statistics, but instead allows for fitting of the full distribution of probabilities of each response type contingent on each stimulus type.

Given that the model cannot meaningfully represent the physical contrast presented to a human participant, we simulated all possible contrast evidence values on a scale of 0 to 5 in steps of 0.2, and calculated the percent correct orientation discrimination performance predicted by the model at each of these contrast evidence levels. We then matched each participant's data to the closest 'contrast evidence' level in the model by identifying 'contrast' levels at which each model performed similarly to the participant on the orientation discrimination task, i.e. matching the percent correct orientation discrimination performance between the model and the data for each contrast level presented to a human participant. The remainder of the summary statistics used in the Behavioral Results analyses – percent of time TP interval was chosen, Type 2 Hit Rate and False Alarm Rate, and percent correct orientation discrimination conditional on TP chosen or unchosen – can then all be derived from this full behavioral profile of stimulus-contingent response probabilities, allowing for comprehensive comparison between model predictions and behavioral data.

We used this measure both to explore the goodness of fit (see below) of the model given no free parameters, and also to evaluate the ideal observer's dependence on the free parameter $\theta$ to the observed behavioral data in all experiments on a subject-by-subject basis, by maximizing the likelihood of the model parameters given the data (**Equation A4-2**) for each model-subject combination. To compare between two models with unequal numbers of parameters, it is standard to use the Bayesian information criterion (BIC), which penalizes more complicated models according to the number of free parameters to avoid overfitting. The BIC is calculated as

$$BIC = -2 \cdot \ln(L_m(\emptyset|data)) + k \cdot \ln(n)$$

$$(A4-3)$$

where $L_m(\emptyset|data)$ is defined as in **Equation A4-2**, $k$ is the number of free parameters, and $n$ is the number of data points.

Maximum likelihood fits were accomplished using a customized Nelder-Mead simplex search algorithm (**Lagarias et al., 1998**) to minimize the negative log-likelihood (**Equation A4-2**). Model predictions were accomplished through Monte Carlo simulation, with 10,000 paired-interval 'trials' at each simulated stimulus contrast level.

## Goodness of fit

To quantify the goodness of fit for the parameter-free ideal observer model presented in the main text, as well as for the model utilizing the best-fitting value of $\theta$ for each participant, we calculated the generalized coefficient of determination, $R^2$, as described in (**Nagelkerke, 1991**). This provides a continuous measure ranging from zero (corresponding to worst fit) to one (corresponding to best fit) based on the maximum likelihood criteria of fit. The formula used is

$$R^2 = 1 - \exp\left[-\frac{2}{n}\left\{l\left(\hat{\beta}\right) - l(0)\right\}\right]$$

(A4 − 4)

where $n$ is the number of data points, and $l(\beta)$ and $l(0)$ denote the log-likelihoods of the fitted and the null model, respectively. We explored two null models: (1) A model which produces every response type with equal probability (i.e., randomly guesses); and (2) a model which knows the correct answer with 100% probability. The generalized $R^2$ is interpreted as the proportion of variance in the data that is explained by the model.

## Biological plausibility

We also considered several other modifications to make our model more biologically plausible. Multiplicative noise is considered due to the observation that neurons often display constant or near-constant Fano factor $F$, where $F = \frac{\sigma^2 W}{\mu_W}$, where $\sigma^2 W$ is the variance and $\mu_W$ the mean of the neuron's firing rates in some time window $W$. In other words, the variance increases as mean firing increases, leading to increased neuronal variability with stronger stimulus inputs. Additionally, both multiplicative and additive noise are considered elements of the noise in neural firing rates and behavioral responses (e.g., **Dosher and Lu, 1998**).

We thus considered two additional free parameters, individually and in conjunction with one another:

*1. f* - signal-dependent (multiplicative) noise, or correction for near-constant Fano factor: the variance-covariance matrix is modified to be $\sum = \begin{bmatrix} cf & 0 \\ 0 & cf \end{bmatrix}$.

*2. $\varepsilon$* - additive noise: the variance-covariance matrix is modified to be $\sum = \begin{bmatrix} \varepsilon & 0 \\ 0 & \varepsilon \end{bmatrix}$

Notably, neither of these parameters – individually or in conjunction with one another – produced any significant effect on qualitative model trends or fit (**Equation A4-2,A4-3**).

## Appendix 5: Alternative models

For completeness, we detail two other models considered here.

### Alternative model 1: heuristic – hierarchical inference

To evaluate whether the behavioral pattern predicted by the Bayesian ideal observer is idiosyncratic or robust to different decision strategies, we explored an alternative model which does not 'ignore' (marginalize over) contrast as a secondary or nuisance variable. Instead, this Bayesian heuristic observer engages in hierarchical inference: it first estimates the most likely contrast to have produced the current stimulus as a secondary or hidden variable, then uses it to inform the decision about the primary variable of interest, that is, the most likely orientation of the stimulus (**Yuille et al., 1996**).

Upon seeing a sample $d$, this observer first makes a guess $\hat{c}_i$ at the most probable $c$ to have generated the current data point $d$, along both the $c_{left}$ and $c_{right}$ dimensions. It does so via maximizing the likelihood of each data point $d$ given the possible source distributions centered at all possible contrast values $c$ along the axes:

$$\hat{c}_{left} = \underset{c_{left}}{argmax}\; p\left(d\middle|\hat{S}_{c_{left}}\right)$$

and

$$\hat{c}_{right} = \underset{c_{right}}{argmax}\; p\left(d\middle|\hat{S}_{c_{right}}\right) \qquad (A5-1)$$

where $\hat{S}_{c_{left}} \sim N\left(\left[c_{left}, 0\right], \sum\right)$ and $\hat{S}_{c_{right}} \sim N\left(\left[0, c_{right}\right], \sum\right)$. This can easily be calculated by a projection of each $d = \left[d_{left}, d_{right}\right]$ onto the $c_{left}$ and $c_{right}$ axes, that is $\hat{c}_{left} = d_{left}$ and $\hat{c}_{right} = d_{right}$ (**Appendix 5—Figure 1**). In other words, the model decomposes the stimulus in each interval into the most likely 'left tilt contrast' ($c_{left}$) and 'right tilt contrast' ($c_{right}$) to have given rise to what the model is 'seeing'. Values for $\hat{c}_{left}$ and $\hat{c}_{right}$ are not constrained to be positive; negative values represent contrast evidence for a left or right tilt that falls below the 'medium gray' of the 'nothing' distribution. Note that one could also assess this for values of $\theta$ other than 90°.

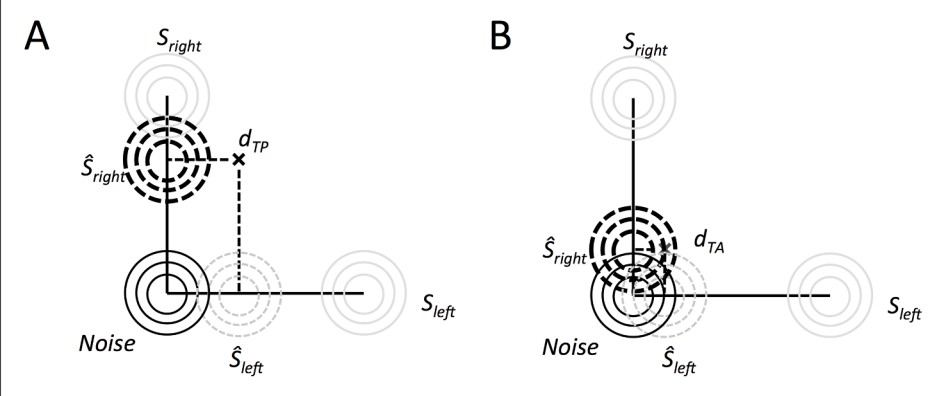

**Appendix 5—Figure 1.** Illustration of the hierarchical inference process.
The observer first makes a best guess about the most likely contrast to have given rise to the evidence it is seeing, and then uses that inferred contrast evidence level to conduct the rest of

the inference process. It does this for both the target-present interval (**A**) and the target-absent interval (**B**).

Once the most likely values for $\hat{c}_{left}$ and $\hat{c}_{right}$ have been obtained, the observer refers to these new 'best guess' source distributions as $\hat{S}_{\hat{c}_{left}}$ and $\hat{S}_{\hat{c}_{right}}$ (**Appendix 5—Figure 1**). The Bayesian heuristic observer determines the posterior probability of each stimulus orientation source distribution $\widehat{S}_{\hat{c}_i}$ for each data sample $d$ via Bayes' Rule,

$$p(\hat{S}_{\hat{c}_I}|d) = \frac{p(d|\hat{S}_{\hat{c}_I})p(\hat{S}_{\hat{c}_I})}{p(d)} \tag{A5-2}$$

with each orientation $\hat{S}_{\hat{c}_I}$ having equal prior probability of 0.5. The observer then makes its orientation decision via

$$\hat{S}_{chosen} = \arg\max_i p(\hat{S}_{\hat{c}_I}|d) \tag{A5-3}$$

To determine which choice the observer is more confident in, the model refers to the magnitude of the posterior probabilities of each $\hat{S}_{chosen}$ in each interval as a measure of the probability of having made a correct orientation discrimination choice, i.e. $p(correct) = p(\hat{S}_{chosen}|d)$. Then the observer compares these posterior probabilities for the Target Present (TP) and Target Absent (TA) intervals by computing the decision variable $D$ via

$$D = \log\left(\frac{p(\hat{S}_{chosen,TP}|d_{TP})}{p(\hat{S}_{chosen,TA}|d_{TA})}\right) \tag{A5-4}$$

As before, if this decision variable is greater than 0, the observer selects the TP interval; if it is smaller than 0, the observer selects the TA interval as being more confident. This model produces the same behavior as the Bayesian ideal observer described in the main text (Appendix 5—Figure 2, top row).

## Alternative model 2: heuristic – likelihood comparison

*Barthelmé et al., 2009* used a variant on the two-by-two forced-choice paradigm (*Nachmias and Weber, 1975*), in which *two* threshold-level stimuli are simultaneously rather than sequentially presented. The observer is asked to choose which stimulus he would be more confident discriminating, and then to discriminate that stimulus. Importantly, in their task, templates are provided on-screen, such that observers may match the stimulus to a provided template rather than a remembered or inferred stimulus dimension.

In that study, the authors compare the performance of a series of models in predicting discrimination and confidence decision, including a Bayesian posterior for each template given the stimulus, and a Bayesian likelihood of each stimulus given each template. The authors report that the likelihood model fits their data better than the posterior model, and conclude that the likelihood method of evaluating confidence describes how human observers judge confidence (despite it being suboptimal). Here, we evaluate the performance of a similar likelihood-only model, defining the decision variable with reference to the most likely contrast level to have given rise to the current sample, i.e.

$$D = log\left(\frac{p(d_{TP}|\hat{S}_{chosen,TP})}{p(d_{TA}|\hat{S}_{chosen,TA})}\right) \tag{A5 − 5}$$

This decision variable is evaluated as described for *Equation A5-5* in the main text. This model failed to produce behavior consistent with human observers' responses (*Appendix 5—Figure 2*, bottom row).

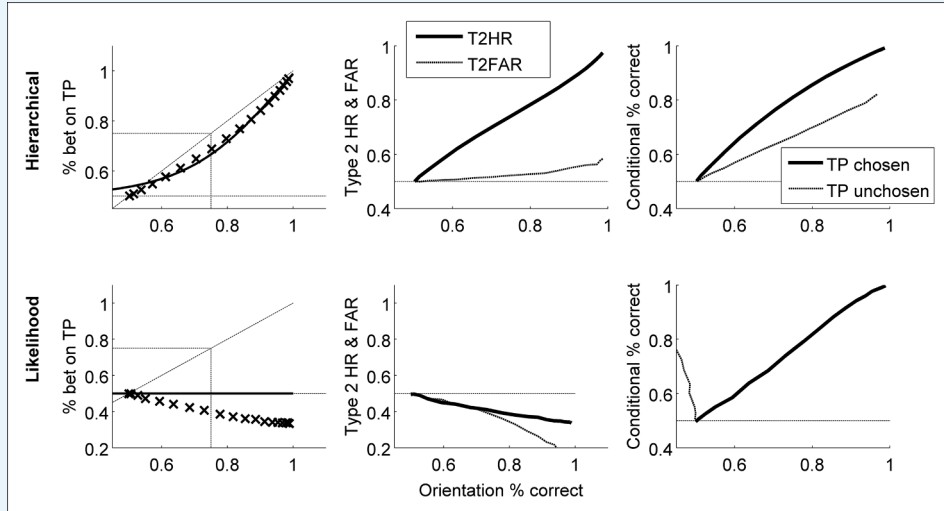

**Appendix 5—Figure 2.** Sample of predicted behavior from Alternative Models 1 (Heuristic - hierarchical Inference) and 2 (Heuristic - likelihood comparison) (at $\theta$ = 90°; see above). While the hierarchical observer produces behavior similar to the ideal observer, the likelihood comparison strategy does not.

