## [Decision Letter]

Thank you for submitting your work entitled "Human observers have optimal introspective access to perceptual processes even for visually masked stimuli" for peer review at *eLife*. Your submission has been favorably evaluated by Timothy Behrens (Senior Editor) and three reviewers, one of whom, Matteo Carandini, is a member of our Board of Reviewing Editors.

The reviewers have discussed the reviews with one another and the Reviewing Editor has drafted this decision to help you prepare a revised submission.

The following individuals responsible for the peer review of your submission have agreed to reveal their identity: Matteo Carandini (Reviewing Editor and Reviewer 3) and David Burr (Reviewer 1). Two other reviewers remain anonymous.

This article has the potential to be important and well cited. Methodologically, it is one of the very best articles that tackle this question of "perception without awareness". Using a new objective technique by Pascal Mamassian the authors argue that there is no evidence for "blindsight" - above-chance discrimination without awareness - in typical healthy adults. The application of appropriate psychophysical methods/ Bayesian modelling is refreshing in a (sub-)field that has often relied on ad-hoc assumptions about how visual signals are encoded and converted to a decision. However, at present the quality of the data are not sufficient to allow the reader to draw unambiguous conclusions. For this paper to have the importance it deserves, it has to have better data. This data should be easy to obtain: it should be straightforward to run the tests on more subjects.

Essential revisions:

1) The main finding rests on the result that participants wagered on the "signal present" interval above chance even when discrimination was at 51%. For this statement to be believable, one needs extra solid statistics. This should be achieved through truly independent samples, i.e. more subjects, not through resampling and statistical values reported to 4 digits of accuracy. Using 3 subjects may be the norm in psychophysics, but here the statistics are essential for the main message and for the claim to be believable one would want easily twice as many. These are easy and cheap measurements so it is not clear why one shouldn't expect a large number of subjects. As well as collecting a couple more subjects, one would also want individual data to be displayed, perhaps as a scatterplot of orientation vs confidence. At the moment, the variability in the current data is such that it's hard to tell whether or not there is a point where discrimination grows but confidence remains flat. Once new data are acquired, this should become a point that can be clearly judged by eye.

2) The paper first establishes by survey that most neuroscientists believe that perception can occur without awareness. This part of the study is amusing but it is unlikely to have lasting value. The figure can easily be replaced by words, and the whole thing dealt with in the Introduction as a motivator for the main part of the paper.

3) The paper should be made more interesting and understandable to general readers, avoiding or at least defining jargon (such as "Type 2 hit rate") and dropping the staid structure of psychophysics papers that divide results as a sequence of experiments.

[Editors' note: further revisions were requested prior to acceptance, as described below.]

Thank you for resubmitting your work entitled "Human observers have optimal introspective access to perceptual processes even for visually masked stimuli" for further consideration at *eLife*. Your revised article has been favorably evaluated by Timothy Behrens (Senior Editor) and a Reviewing Editor, Matteo Carandini. The manuscript has been markedly improved but there are some remaining minor issues that need to be addressed before acceptance. These minor issues all concern the text, which describes the material in a way that is sometimes rushed and garbled. They can all be solved with a simple round of editing.

1) A useful rule of thumb is to describe the figures one by one, without relying on people having to read captions or future parts of the paper or Methods.

2) It is premature to point to Figure 1 and Figure 2 in Introduction. To read them and digest them is too much to ask to a reader who is still in Introduction, and has had no text to explain those two figures. That's what Results is for.

3) The Results section seems in a hurry to give take-home messages, without taking the time explain the figures. Please devote at least a paragraph to a description of Figure 1 (not zero words, as currently in Line 109 the first line of subsection “Behavioral Experiments”), guiding the reader through it. Similarly, please devote at least a paragraph to a description of Figure 2, and guide the reader through it.

4) Related to the previous points, please move much of the material from captions to main text. Captions should be used to explain what is in the graphs. They can also be used to described results and take-home messages if desired, but this is not a requirement, and that material should certainly also be in the main text.

5) It is not advisable to use main text to refer the reader to a figure caption (L119 last line of first paragraph subsection “Behavioral Experiments”), especially when that caption in turn contains another pointer, to another part of Results and to another caption (L152-153 Figure 2 legend). Please unravel all that material.

6) Figure 2 still has some jargon, e.g. the word "Metacognitive", which is not explained anywhere in the paper. Also, "Type 2 hit rate" remains mysterious to a non-expert reader. Is there a Type 1 hit rate? How about Type 3? What is a Type? Please define in text or do not use.

7) Subsection “Behavioral Experiments” abruptly refers to "Experiment 2" and "Experiment 1", as if the readers already knew that there are two experiments. Explain that there are two experiments, called "1" and "2", explain their differences in design, and then explain that you saw no difference in the results, pointing to appropriate panels in Figure 3. In fact, all this can be done after having described Figure 1 and Figure 2: the logical place seems to be when describing Figure 3.

8) Perhaps consider whether it is really a good idea to anticipate the take-home message of the paper on the 8th line of Results (Line 115)? This adds to the feeling that the paper is rushing to give a result without really explaining much. Perhaps it is ok to do this after the paper has been edited and Figure 1 and Figure 2 have been properly described.

9) This list of suggestions ends here but it would be good to look at the whole paper for clarity and readability. The authors may be too immersed in the results and in the prose to be the best judges of this, so asking a naive colleague might help.

---

## [Author Response]

*Essential revisions: 1) The main finding rests on the result that participants wagered on the "signal present" interval above chance even when discrimination was at 51%. For this statement to be believable, one needs extra solid statistics. This should be achieved through truly independent samples, i.e. more subjects, not through resampling and statistical values reported to 4 digits of accuracy. Using 3 subjects may be the norm in psychophysics, but here the statistics are essential for the main message and for the claim to be believable one would want easily twice as many. These are easy and cheap measurements so it is not clear why one shouldn't expect a large number of subjects. As well as collecting a couple more subjects, one would also want individual data to be displayed, perhaps as a scatterplot of orientation vs confidence. At the moment, the variability in the current data is such that it's hard to tell whether or not there is a point where discrimination grows but confidence remains flat. Once new data are acquired, this should become a point that can be clearly judged by eye.*

We agree. We have collected an additional 6 subjects’ worth of data, making 9 total subjects in Experiment 1; we opted to focus on Experiment 1, as Dr. Carandini (Reviewer 3) pointed out that Experiment 2 is essentially a replication and does not necessarily need its own section anyway. Instead of using bootstrapping and t-tests of the fitted psychometric functions (see also our response to Dr. Carandini’s comment about Figure 4), we rely on quantitative goodness of fit metrics between a version of the Bayesian observer modified to produce sub-optimal Performance without Awareness to demonstrate that the ideal observer (which of course does not produce Performance without Awareness) provides the best fit to the human data. We also overlay the group mean data on a scatterplot of individual subjects’ orientation percent correct versus betting on the target-present interval as suggested, to allow the reader to easily see that the pattern of responses in no way resembles Performance without Awareness. This is an excellent suggestion for visualization of the data that really drives home the message. We have also included the individual subjects’ data in Appendix 2, as before.

*2) The paper first establishes by survey that most neuroscientists believe that perception can occur without awareness. This part of the study is amusing but it is unlikely to have lasting value. The figure can easily be replaced by words, and the whole thing dealt with in Introduction as a motivator for the main part of the paper.*

Agreed. We have done as suggested, and moved the details of the survey study and its results to Appendix 1. We now include the following text as a summary of the survey in the Introduction:

“We conducted an informal survey to confirm this popular belief, which also revealed that convincing evidence for this phenomenon is believed to be lacking. We asked survey participants three key questions: (1) “Do you believe in subliminal perception?” (2) “Do you believe that the subjective threshold for awareness is above the objective discrimination threshold?” and (3) “If ‘yes’, do you believe this has been convincingly demonstrated in the literature?” Most respondents reported believing that subliminal processing exists (94%), but also that they did not believe it had been convincingly demonstrated in the literature (64%). These belief patterns were shown even among those who reported having published on subliminal or unconscious perception (94% and 61%, respectively). See Appendix 1 for full text of questions and detailed survey results.”

*3) The paper should be made more interesting and understandable to general readers, avoiding or at least defining jargon (such as "Type 2 hit rate") and dropping the staid structure of psychophysics papers that divide results as a sequence of experiments.*

Also agreed. We have removed all but the most essential abbreviations throughout the text, opting instead to spell out the terms in words. We also now accompany any necessary jargon-y phrases (e.g. Type 2 hit rate) with definitions. To help the paper flow better, we have also combined the results from Experiments 1 and 2 into a single Behavioral Experiments results section, in response to the point that Experiment 2 is basically a replication of Experiment 1. The new text reads:

“Because results are very similar across the two experiments, we combined results from both and performed a two-tailed one-sample t-test to assess whether this predicted percentage betting on the target-present interval significantly diverged from 75%. This analysis revealed that observers bet on the target-present interval significantly less than 75% of the time at 75% correct orientation discrimination accuracy (Figure 3, Table 1). Thus, observers exhibited Performance > Awareness (see also Modeling Results, below).”

We have also removed the long lists of statistics (as we changed the statistical tests used, see specific comments below), and replaced them with tables where appropriate.

To clarify the section of text that contained the most jargon before, we have placed an additional heading of “2IFC detection?” in the results section, to help the reader understand what is being discussed. In this section, the modified text reads (in context):

“To confirm that subjects were indeed rating confidence, we plotted Type 2 hit rate (placing a bet on a correct orientation discrimination decision) and Type 2 false alarm rate (placing a bet on an incorrect orientation discrimination decision) against orientation discrimination accuracy (Figure 3). Subjects displayed increasing Type 2 hit rate as a function of orientation discrimination accuracy, whereas Type 2 false alarm rate remained relatively flat at around 50% (chance level) across increasing orientation discrimination accuracy.”

In the Bayesian Ideal Observer Model section, we also have removed acronyms. The new text reads:

“Crucially, however, the ideal observer does exhibit Performance > Awareness (Figure 3), and to a similar extent as our human participants (R2 = 0.565; see Appendix 4 for details of goodness of fit metrics); trends for Type 2 hit and false alarm rates (Figure 3), and percent correct conditional upon having bet on the target-present versus target-absent interval (Figure 3), also match human data.”

[Editors' note: further revisions were requested prior to acceptance, as described below.]

*1) A useful rule of thumb is to describe the figures one by one, without relying on people having to read captions or future parts of the paper or Methods.*

Thank you for this suggestion. We have now moved the majority of information from the captions into the main text, and/or expanded the information previously contained in the captions in the main text as well.

*2) It is premature to point to Figure 1 and Figure 2 in Introduction. To read them and digest them is too much to ask to a reader who is still in Introduction, and has had no text to explain those two figures.*

Agreed. We have now removed references to Figure 1 and Figure 2 in the Introduction.

*3) The Results section seems in a hurry to give take-home messages, without taking the time explain the figures. Please devote at least a paragraph to a description of Figure 1 (not zero words, as currently in the first line of subsection “Behavioral Experiments”), guiding the reader through it. Similarly, please devote at least a paragraph to a description of Figure 2, and guide the reader through it.*

Agreed. At the beginning of the Results section we have now explained the methods in more detail to give the reader some context, and also devoted a paragraph each to describing both Figure 1 and Figure 2. Here we also introduce the two experiments, since referring to Figure 1 means the reader will be introduced to there being two experiments at this juncture.

The new text at the beginning of the Results section reads:

“Nine human observers participated in two experiments of our 2IFC confidence-rating paradigm (Figure 1). In both experiments, participants viewed two intervals in which they were required to discriminate the orientation (right or left tilt) a Gabor patch target embedded in forward- and backward-masks (Figure 1), and judged which of the discrimination choices they felt more confident in. Crucially, in one of the intervals the target was absent (Figure 1), such that above-chance discrimination performance was impossible. We performed two experiments to assess the potential contributions of question order, receipt of feedback, and a priori knowledge of the presence of a target-absent interval (Figure 1). In Experiment 1, participants judged which decision they felt more confident in and then indicated their orientation decisions for both intervals, while in Experiment 2 they indicated their orientation discrimination decisions before selecting the more-confident interval. In Experiment 2, we also provided feedback on the confidence decision, and told participants that one interval contained no target; this information was withheld from participants in Experiment 1. Stimuli, timing details, and order of question prompts in the two experiments are also discussed in greater detail in the Methods section.

For both experiments, we evaluated whether participants exhibited Performance without Awareness (Figure 2) or Performance > Awareness (Figure 2). In both cases, the response pattern of interest can be visualized as percent of time betting on the target-present interval as a function of percent correct orientation discrimination in the target-present interval. ‘Performance without Awareness’ (Figure 2) would be supported if observers can discriminate the target above chance (>50% accuracy) while being unable to bet on their choices more often than betting on the target-absent interval (which necessarily yields chance-level performance). That is, observers correctly discriminate the target’s orientation more than 50% of the time, but bet on the target-present interval 50% of the time (i.e., they bet randomly on the target-present versus target-absent interval), indicating they are not aware of the information that contributed to their discrimination decision. If this were to occur, it would most likely happen at low discrimination performance levels, yielding a pattern of behavior similar to that presented in Figure 2.

However, in psychophysics, thresholds can also be defined as midway between ceiling and floor performance (Macmillan & Creelman, 2004), such that threshold discrimination performance is defined as 75% accuracy rather than >50% (chance level). This concept can also be applied to subjective betting data in the sense that betting on the target-present interval could be considered “correct” or “advantageous” betting. In this sense (threshold = 75% correct performance), the subjective threshold for confidence might be above the objective threshold for discrimination. In other words, observers may bet on the target-present interval less often than they get the discrimination correct, but still above chance. This would occur because the orientation discrimination choice requires evaluation of only one interval (the one with the target in it) and therefore is subject to only one source of uncertainty, but the “betting” choice requires evaluation of both intervals, and therefore has two potential sources of uncertainty. This pattern of behavior (Figure 2) may occur even if subjects do not display Performance without Awareness, and would be characterized by a pattern of responses that fall below the identity line (diagonal dashed line). We call this possibility ‘Performance > Awareness’.”

*4) Related to the previous points, please move much of the material from captions to main text. Captions should be used to explain what is in the graphs. They can also be used to described results and take-home messages if desired, but this is not a requirement.*

Agreed. Done.

*5) It is not advisable to use main text to refer the reader to a figure caption (last line of first paragraph subsection “Behavioral Experiments”), especially when that caption in turn contains another pointer, to another part of Results and to another caption (Figure 2 legend). Please unravel all that material.*

Done. The new portion of the main text contains the information previously included in a caption with some additional explanation. Please see our response to Comment 3 for the new text.

*6) Figure 2 still has some jargon, e.g. the word "Metacognitive", which is not explained anywhere in the paper. Also, "Type 2 hit rate" remains mysterious to a non-expert reader. Is there a Type 1 hit rate? How about Type 3? What is a Type? Please define in text or do not use.*

We have significantly shortened the caption to Figure 2, moving most of it to the main text and expanding in greater detail. In the ‘2IFC detection?’ section, we have also now expanded the definition of Type 2 hits and false alarms in the main text, and contrasted them to Type 1 hits and false alarms according to standard signal detection theoretic definitions in order to make the definition clearer to the reader.

The new text in context reads:

“One possible concern is that subjects were not rating confidence but instead engaging in 2IFC detection of the target-present interval. To confirm that subjects were indeed rating confidence, we plotted Type 2 hit rate and Type 2 false alarm rate against orientation discrimination accuracy (Figure 3). A Type 2 hit is defined as placing a bet on a correct orientation discrimination decision, whereas a Type 2 false alarm is defined as placing a bet on an incorrect orientation discrimination decision. These are in contrast to Type 1 hits and false alarms, which can be defined as saying “left” when a left-tilted Gabor was presented and saying “left” when a right-tilted Gabor was presented, according to standard signal detection theoretic definitions (Green & Swets, 1966; Macmillan & Creelman, 2004).”

A brief explanation is also included in the caption for Figure 3, in case readers read the caption without reading the main text. The relevant portion of that caption reads:

“Panels B, E, and H show rising Type 2 hit rate (‘HR’; when subjects bet on a correct orientation discrimination choice) but relatively flat Type 2 false alarm rate (‘FAR’; when subjects bet on an incorrect orientation discrimination choice), …”

We have also added a definition of “metacognitive” to the ‘Unconscious “hunches”?’ section of the Results. That new text now reads:

“However, one concern might be that subjects are able to meaningfully rate confidence despite no subjective visual experience of the stimulus due to some sort of non-visual “hunch” or “feeling.” Indeed, such metacognitive insights (the ability to introspectively distinguish between correct and incorrect responses) have recently been reported even in the absence of objective task performance sensitivity, although not in the context of perception (e.g., Scott, Dienes, Barrett, Bor, & Seth, 2014).”

*7) Subsection “Behavioral Experiments” abruptly refers to "Experiment 2" and "Experiment 1", as if the readers already knew that there are two experiments. Explain that there are two experiments, called "1" and "2", explain their differences in design, and then explain that you saw no difference in the results, pointing to appropriate panels in Figure 3. In fact, all this can be done after having described Figure 1 and Figure 2: the logical place seems to be when describing Figure 3.*

As mentioned in our response to Comment 3, we have now included an expanded description of the two experiments at the beginning of the Results section. We believe the reference to the two experiments now makes sense, and when we discuss analyzing the two experiments together, we point the reader to the relevant panels of Figure 3.

*8) Perhaps consider whether it is really a good idea to anticipate the take-home message of the paper on the 8th line of Results? This adds to the feeling that the paper is rushing to give a result without really explaining much. Perhaps it is ok to do this after the paper has been edited and Figure 1 and Figure 2 have been properly described.*

As you anticipated, with the full descriptions of Figure 1 and Figure 2 now in the main text, as well as a description of the differences between the two experiments, we believe this is okay at this point.

*9) This list of suggestions ends here but it would be good to look at the whole paper for clarity and readability. The authors may be too immersed in the results and in the prose to be the best judges of this, so asking a naive colleague might help.*

We have now had a naive colleague read the paper and make suggestions throughout. We have implemented these suggestions to aid the readability of the paper to non-experts in the field. All changes are tracked. For example, in the Introduction we now unravel the terms “objective” and “subjective” a bit more.

“In these demonstrations (e.g., blindsight: Weiskrantz, 1986) the subjective threshold for awareness (when a stimulus is consciously “seen”) seems well above the objective threshold for forced-choice discrimination (when a stimulus can be correctly identified): subjects can discriminate a target above chance performance, yet report no awareness of the target.”

We also include a definition of 2-interval forced choice (2IFC) in the appropriate spot at the end of the Introduction:

“Subjects discriminated two stimulus intervals, only one of which contained a target, and indicated confidence in their decisions using a 2-interval forced-choice procedure (2IFC), i.e. indicating which of the two discrimination decisions they felt more confident in.”

And to give the reader a little more context for the paper, we include the following statement at the end of the Introduction:

“Here, we explored whether such Performance without Awareness occurs in normal observers in two behavioral experiments, and compared these results to predictions of a Bayesian ideal observer.”

These changes are accompanied by minor insertions and word substitutions throughout. We believe these changes improve the clarity of the manuscript in the manner suggested by the reviewer.